# Molecular basis of Fab-dependent IgA antibody recognition by gut-bacterial metallopeptidases

María Ángeles Márquez-Moñino [1,6], Ana Martínez Gascueña [1,6], Tala Azzam [2], Andrea Persson[3], Aitor Manzanares-Gomez[1], Marina Aguillo-Urarte [1], Trenton T Brown[2], Ainhoa Montero-Sagarminaga[1], Rolf Lood [3], Andreas Naegeli[3], Sean R Connell[4,5], Diego E Sastre [2], Eric J Sundberg[2] & Beatriz Trastoy [1,5✉]

## Abstract

**Immunoglobulin A (IgA) is essential for mucosal immunity and has been implicated in autoimmune diseases, such as IgA nephropathy. Certain pathogenic and commensal bacteria produce IgA proteases that selectively cleave IgA, potentially aiding bacterial colonization as well as suggesting therapeutic avenues for IgA nephropathy. Here, we investigate the substrate specificities of two enzymes of the M64 metallopeptidase family, the IgA protease ThomasA from *Thomasclavelia ramosa* and BF3526 from *Bacteroides fragilis*. Our structural, biochemical, and mutagenesis analyses demonstrate that ThomasA cleaves IgA through exclusive recognition of the Fab region. This mechanism is distinct from that of other antibody-specific peptidases, which typically require engagement of the Fc region. In contrast, X-ray crystal structures of BF3526 in complex with substrate and product peptides, combined with enzymology assays, show that this enzyme targets the N-terminus of pre-digested proteins, but does not act on intact IgA. These findings reveal divergent substrate recognition strategies between M64 family members, while providing new structural insights into their conserved catalytic mechanism.**

**Keywords** Immunoglobulin A; Peptidases; *Thomasclavelia ramosa*; Bacteroides; IgA Nephropathy
**Subject Categories** Immunology; Microbiology, Virology & Host Pathogen Interaction; Structural Biology

## Introduction

Immunoglobulin A (IgA) is the most abundantly produced antibody isotype in the human body and the primary antibody found in secretions such as tears, saliva, respiratory and intestinal fluids, and colostrum (Bunker and Bendelac, 2018; Dingess et al, 2022). It is also the second most prevalent antibody in human serum, with concentrations ranging from 1 to 3 mg/mL (Kerr, 1990). Secretory IgA plays a crucial role in mucosal immunity and microbiota homeostasis (Pabst and Slack, 2020; Mantis et al, 2011), while serum IgA contributes to pro-inflammatory responses by inducing cytokine and chemokine release, promoting phagocytosis and degranulation, and triggering neutrophil extracellular trap formation (Hansen et al, 2019; Steffen et al, 2020; Breedveld and Van Egmond, 2019). Due to these inflammatory effects, dysregulation of serum IgA is implicated in several autoimmune diseases (Breedveld and Van Egmond, 2019), such as inflammatory bowel disease (Lin et al, 2018), rheumatoid arthritis (RA) (Derksen et al, 2022), IgA nephropathy (IgAN) (Suzuki et al, 2009; Coppo and Amore, 2004), and transplant rejection (Arnold et al, 2018).

IgA exists in humans as two subclasses, IgA1 and IgA2. In serum, IgA1 comprises ~90% of total IgA, while its ratio to IgA2 varies across secretions (Kerr, 1990; Brandtzaeg and Johansen, 2005). In addition, IgA2 is found in three allelic forms, IgA2m(1), IgA2m(2), and IgA2m(n) (Tsuzukida et al, 1979; Infante and Putnam, 1979). Despite high sequence similarity (Steffen et al, 2020), IgA1 and IgA2 exhibit distinct glycosylation patterns (Fig. 1A). Both subclasses are heavily glycosylated, with IgA1 containing one *N*-glycosylation site in the Fc region of each heavy chain (HC), and IgA2 having two in the Fc region and an additional site in each Fab region. IgA1 and IgA2 also share a *N*-glycosylation site in the tailpiece region, which plays a role in IgA dimerization (Atkin et al, 1996). Moreover, the hinge region (HR) of IgA1, which is 13 amino acids longer than IgA2's, has three to six *O*-glycans (Reily et al, 2019) (Fig. 1A,B). These *O*-glycans, primarily composed of *N*-acetylgalactosamine (GalNAc), galactose (Gal), and *N*-acetylneuraminic acid (Neu5Ac), exhibit considerable variability in composition and number, contributing to the microheterogeneity of IgA1 molecules (Lehoux et al, 2014). In IgAN patients, *O*-glycans in the HR of IgA1 exhibit a deficiency in Gal and a relative increase in GalNAc. These aberrant glycoforms, known as Gal-deficient IgA1 (Gd-IgA1) (Moldoveanu et al, 2007;

[1]Structural Glycoimmunology Laboratory, Biobizkaia Health Research Institute, Barakaldo, Spain. [2]Department of Biochemistry, Emory University School of Medicine, Atlanta, GA, USA. [3]Genovis AB, Box 790, Lund 22007, Sweden. [4]Structural Biology of Cellular Machines Laboratory, Biobizkaia Health Research Institute, Barakaldo, Spain. [5]Ikerbasque, Basque Foundation for Science, Bilbao 48009, Spain. [6]These authors contributed equally: María Ángeles Márquez-Moñino, Ana Martínez Gascueña. ✉E-mail: beatriz.trastoybello@bio-bizkaia.eus

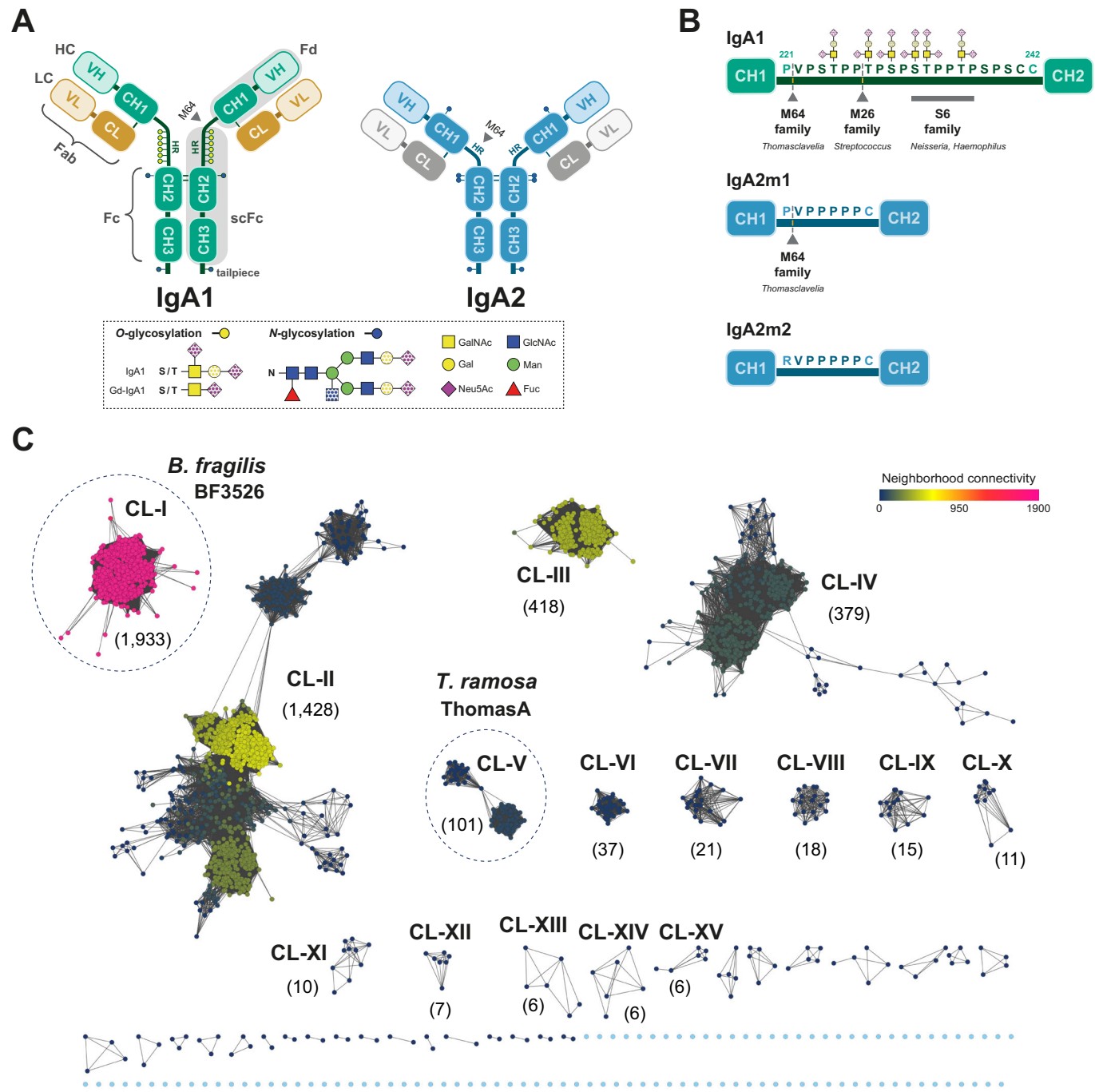

**Figure 1. Schematic representation of IgA1 and IgA2 and Sequence Similarity Network analysis of M64 peptidase family.**

(A) Schematic representation of IgA1 and IgA2m(1), highlighting their distinct domains and *O*- and *N*-glycosylation sites. The dotted shapes indicate that there may or may not be a carbohydrate present, highlighting the considerable structural diversity of the *O*- and *N*-glycans. CH constant heavy chain domain, CL constant light chain domain, HC heavy chain, LC light chain, VH variable heavy chain domain, VL variable light chain domain. (B) Cleavage sites within the hinge region of different IgA1 and IgA2 allotypes targeted by various IgA peptidase families. (C) Sequence similarity network (SSN) analysis of the M64 peptidases, illustrating evolutionary relationships and functional clustering. Each node represents a group of proteins sharing at least 35% sequence identity, using an alignment score threshold of 50. Node size is scaled according to the number of proteins it represents. Nodes were organized with the Cytoscape Prefuse Force Directed Open CL layout % id and colored by neighborhood connectivity using Enzyme Similarity Tool (EFI-EST). Neighborhood connectivity coloring measures local network interconnectivity for the detection of protein families within SSN clusters. Highly interconnected clusters share sequence similarity and potentially functional similarity. Source data are available online for this figure.

Tomana et al, 1999), are associated with altered expression of specific enzymes involved in *O*-glycan biosynthesis (Suzuki et al, 2008).

Several pathogenic and commensal bacterial species that colonize mammalian mucosal membranes produce IgA proteases (IgAPs), which specifically inactivate this antibody isotype through proteolytic cleavage, potentially serving as a key survival mechanism (Shon et al, 2021; Kilian et al, 1983; Kosowska et al, 2002; Lu et al, 2024). Some *Streptococcus* species produce M26 family metalloproteases (Woof and Mestecky, 2005; Wang et al, 2020; Chi et al, 2017), while certain *Haemophilus* and *Neisseria* species produce S6 family serine proteases (Mulks et al, 1980; Chintalacharuvu et al, 2003), according to the MEROPS database (Rawlings et al, 2018). Though most IgAPs originate from pathogens, *Thomasclavelia ramosa* (formerly *Clostridium ramosum*), a commensal human gut microbe, expresses ThomasA (also known as AK183 (Xie et al, 2022), IgA-peptidase of *Clostridium ramosum* (Cerdà-Costa and Gomis-Rüth, 2014), *C. ramosum* IgA proteinase (Kosowska et al, 2002), IgA protease of *Clostridium spp.* (Fujiyama et al, 1985)), a metalloprotease from the M64 family (Kosowska et al, 2002). The substrate specificity of IgAPs varies among families. M26 and S6 are specific for IgA1, whereas M64 hydrolyzes both IgA1 and IgA2. M26 metalloproteases cleave IgA1 at P227–T228 (amino acid numbering follows the commonly adopted convention (Putnam et al, 1979)) within the HR, while S6 serine proteases target multiple hydrolytic sites within the HR, between S232 and P237 (Shon et al, 2021; Plaut et al, 1975) (Fig. 1B). ThomasA, the only characterized M64 family member, cleaves both IgA1 and IgA2 at P221–V222, just upstream of the HR in IgA1 and IgA2m(1) (Kosowska et al, 2002). However, IgA2m(2) is resistant to cleavage due to a P221R substitution (Kosowska et al, 2002). Despite containing a metalloprotease domain with a putative zinc-binding motif, ThomasA shares little sequence homology with M26 proteases. This multidomain protein (~130 kDa) harbors a zinc-binding **HEXXHXXXGXXD** motif within its predicted catalytic domain, a hallmark of zinc-dependent metalloproteinases (Kosowska et al, 2002). However, the role of the additional domains in ThomasA and its mechanism of interaction with IgAs remain poorly understood, limiting our ability to define its contribution to bacterial colonization in the human gut microbiome.

Recently, due to their remarkable specificity for IgA antibodies, IgAPs have been explored as potential treatments for IgAN (Xie et al, 2022; Lamm et al, 2008; Lechner et al, 2016), an autoimmune disease for which no specific therapy is currently available. IgAN is the most prevalent primary glomerular disease worldwide, with a variable prognosis; 20–40% of patients develop end-stage renal disease requiring dialysis or transplantation within 20 years of diagnosis (Floege and Amann, 2016; Coppo and D'Amico, 2005). This disease is characterized by immune complex deposition in the kidney, consisting of Gd-IgA1, complement proteins, and auto-antibodies that specifically recognize the aberrantly glycosylated IgA1 HR (Coppo and Amore, 2004). These immune complexes interact with receptor cells in the glomeruli and other complement proteins, triggering inflammation and kidney damage (Floege et al, 2014). Due to the ability of IgAPs to target IgA1 and dissolve these pathogenic immune complexes, several preclinical studies have been performed as proof-of-concept with recombinant IgAPs. Notably, *Haemophilus influenzae* IgA1-P from the S6 family

significantly reduced human IgA1 mesangial deposits, inflammation, fibrosis, and hematuria in a mouse IgAN model (Lechner et al, 2016). Additionally, a preliminary in vivo study in an IgAN mouse model used a chimeric Fc-fused ThomasA, which exhibited prolonged efficacy, highlighting its therapeutic potential in humans (Xie et al, 2022). Its origin from a commensal bacterium may also confer the added advantage of low immunogenicity (Xie et al, 2022). However, the molecular mechanisms that govern IgA specificity by ThomasA remain unknown, limiting its full therapeutic potential and hindering the development of novel enzymatic strategies for IgAN.

In this study, we investigate the catalytic and molecular mechanisms of substrate recognition by M64 metallopeptidases, considering their potential involvement in bacterial colonization of the gut and their possible application in IgAN treatment. While ThomasA remains the only characterized M64 family member, the molecular basis that defines its substrate specificity, domain contributions to IgA recognition, and broader functional diversity within the M64 family are still poorly understood. To address this, we employ structural biology, small-angle X-ray scattering (SAXS), alanine scanning mutagenesis, and enzymology to dissect the molecular interactions that mediate IgA recognition by ThomasA. In addition, we use bioinformatics, liquid chromatography-mass spectrometry (LC-MS) and X-ray crystallography to uncover novel specificities within other family members, such as BF9343_3433, also known as BF3526, from the human gut bacterium *Bacteroides fragilis*, and to understand its catalytic mechanism. This comprehensive analysis reveals key structural determinants of substrate specificity and enzyme function, shedding light on how ThomasA engages with IgA and how other M64 peptidases exhibit distinct substrate preferences. These findings not only advance our understanding of IgA recognition and cleavage by bacterial proteases but also provide a framework for harnessing these enzymes in biotechnological and therapeutic applications.

## Results

### Sequence variability of the M64 peptidase family

In order to define sequence-structure-function relationships within the metallopeptidase M64 family, we performed a sequence similarity network (SSN) analysis, employing all the M64 family members contained in the Pfam families PF16217 and PF09471, which are also included in the MEROPS database (Fig. 1C). Based on our SSN analysis, these enzymes can be classified into 15 main clusters, each containing at least six proteins. Highly interconnected clusters are expected to share sequence and, potentially, functional similarities. The largest cluster, CL-I, comprises 1933 proteins and exhibits the highest connectivity among all clusters (neighborhood connectivity (NC) range of 1547–1905 and a convergence ratio = 0.94) (Fig. 1C). Notably, more than 60% of CL-I members belong to the phylum Bacteroidota, which includes diverse genera such as *Bacteroides*, *Alistipes*, *Prevotella*, and *Capnocytophaga*. These bacteria commonly colonize various human body sites, including the skin, oral cavity, urogenital tract, and, predominantly, the gastrointestinal tract, forming part of the human gut microbiota. The remaining members of CL-I belong to the phyla Acidobacteriota, Ignavibacteriota, Myxococcota, and

Calditrichota. The second-largest cluster, CL-II, contains 1,428 proteins and is one of the most taxonomically diverse groups. Approximately half of its members belong to the class *Actinomycetia*, while representatives from the phyla Bacteroidota, Pseudomonadota, and, to a lesser extent, Acidobacteriota, Bacillota, and Verrucomicrobiota are also present. CL-III, consisting of 418 proteins, is another highly interconnected cluster (NC = 210–403), with 80% of its members belonging to the fungal kingdom, including species from the phyla Basidiomycetota, Oomycetota, and Ascomycetota. CL-IV and CL-XII are exclusively composed of members of the phylum Bacteroidota (class *Bacteroidia*), with 379 and 7 representatives, respectively. CL-V consists of 101 proteins from the phylum Bacillota (mostly composed of orders *Eubacteriales* and *Lachnospirales*) and is primarily constituted of metallopeptidases from human gut bacteria, including *Roseburia intestinalis*, *Agathobacter rectalis*, *Eubacterium ventriosum*, *Coprococcus* sp., *Lachnospira eligens*, *Clostridium* sp., and the former *Clostridium* species *Thomasclavelia ramosa* (order *Erysipelotrichales*), which encodes the IgAP metallopeptidase ThomasA (Kosowska et al, 2002). The analysis revealed that CL-V members exhibit relatively high sequence variability (34–99% sequence identity range within each protein in the cluster) with a convergence ratio = 0.62 and a neighborhood connectivity range of 24–75. Almost all nodes of CL-VI, CL-VII and CL-XIII contain representatives of phylum Pseudomonodota, while CL-VIII, CL-IX and CL-XIV are composed of members of phylum Planctomycetota. Despite this variability, all proteins across clusters retain the characteristic catalytic motif **HEXXHXXXGXXD**, with some variations in CL-III and CL-VI where the conserved Asp residue is replaced by Glu (Appendix Fig. S1). Notably, CL-X, which consists exclusively of 11 members of the *Trypanosomatidae* family does not contain the conserved catalytic residues characteristic of the M64 family, despite being included in Pfam 09471.

## IgAP ThomasA from gut microbiome is a monomeric multidomain protein

ThomasA (UniProt code Q9AES2) is a 137.2 kDa secreted protein with a signal peptide (SP) sequence (residues 1–30) and a C-terminal SPXTG motif (residues 1196–1200), a potential sortase recognition site that likely contributes to cell wall anchoring (Kosowska et al, 2002). The AlphaFold (AF) (Jumper et al, 2021) model of ThomasA (AF-Q9AES2-F1), excluding the SP and the disordered C-terminal region (ThomasA$^{31-1167}$), reveals at least eight domains from the N- to C-terminus: a N-terminal domain of unknown function DUF6273 (NTD, residues 31–322), an M64 peptidase domain (residues 323–632), and a set of three β−sandwich (β1,2,5) and three β−sheet domains (β3,4,6 domains) (residues 633–1167) (Fig. 2A). A structural homology search using the DALI server (Holm et al, 2023) identified peptidases with significant structural similarity to the M64 peptidase domain of ThomasA (Appendix Fig. S2 and Appendix Text). The closest homologs include the M64 peptidase from *Bacteroides fragilis* strain NCTC 9343, BF3526 (PDB code 4DF9; Z-score of 25.5; root mean squared deviation (r.m.s.d.) of 2.3 Å for 234 aligned residues; 21% identity) and the M64 peptidase from *B. ovatus*, BACOVA_00663 (PDB code 3P1V; Z-score of 22.7, r.m.s.d. of 2.5 Å for 233 aligned residues; 22% identity). Notably, BF3526 and BACOVA_00663 belong to cluster CL-I in our SSN analysis but have not yet been

biochemically characterized. The remaining ThomasA domains exhibited low or no structural similarities to reported protein structures (Appendix Fig. S2 and Appendix Text).

To investigate the overall structure of ThomasA in solution and the spatial arrangement of its domains, we performed in-line size-exclusion chromatography SAXS (SEC-SAXS; Fig. 2; Appendix Tables S1 and S2; Appendix Fig. S3). We analyzed ThomasA$^{31-1167}$, along with two truncated variants: ThomasA$^{31-878}$, which lacks the β3–β6 domains, and ThomasA$^{323-878}$, which excludes the NTD and the β3–β6 domains (Fig. 2A; Appendix Tables S1 and S2). Attempts to purify ThomasA$^{31-795}$, ThomasA$^{31-632}$ and ThomasA$^{323-632}$ constructs were unsuccessful due to a lack of expression. The three successfully expressed constructs eluted from the size exclusion chromatography column as monomers, with average molecular weights of 155.1, 91.7, and 69.2 kDa, respectively, calculated from experimental SAXS data (Appendix Table S2). SAXS analysis determined the radius of gyration ($R_g$) to be 64.4, 34.4, and 29.5 Å for ThomasA$^{31-1167}$, ThomasA$^{31-878}$, and ThomasA$^{323-878}$, respectively (Fig. 2B–D; Appendix Table S2). The interatomic distribution function $P(r)$ profile of ThomasA$^{31-1167}$ exhibited a main peak at an r-value of ~38 Å with a long tail extending to a maximum dimension ($D_{max}$) of 250 Å, suggesting a highly elongated conformation (Receveur-Brechot and Durand, 2012) (Fig. 2C). In contrast, the monomodal $P(r)$ functions of constructs ThomasA$^{31-878}$ and ThomasA$^{323-878}$ showed $D_{max}$ values of 106 and 92 Å, respectively (Fig. 2C; Appendix Table S2). The normalized Kratky plot revealed a bell-shaped profile for ThomasA$^{31-878}$ and ThomasA$^{323-878}$, indicative of a compact, globular structure (Fig. 2D). In contrast, the Kratky plot for ThomasA$^{31-1167}$ exhibited a combination of the bell shape and a plateau that gradually decays to zero, suggesting that the protein contains flexible regions (Receveur-Brechot and Durand, 2012), likely originating from the C-terminal β3-β6 domains (Fig. 2D).

We reconstructed the ab initio low-resolution envelopes of ThomasA$^{31-1167}$, ThomasA$^{31-878}$ and ThomasA$^{323-878}$ using GASBOR (Fig. 2E–G). The SAXS reconstruction reveals that ThomasA$^{323-878}$ and ThomasA$^{31-878}$ adopt asymmetric bean-like and kidney-shaped conformations, respectively, with their AlphaFold 3 (AF3) (Abramson et al, 2024) models fitting into the ab initio envelopes (Fig. 2E,F; Appendix Fig. S3). The kidney-shaped arrangement of the NTD, M64 peptidase and β1–β2 domains in the ThomasA$^{31-878}$ was also confirmed by electron microscopy (EM), where a 3D reconstruction of negatively stained particles yields a map consistent with the SAXS and AF3 models (Appendix Fig. S4). The SAXS envelope of ThomasA$^{31-1167}$ exhibits an extended conformation, where the NTD, M64 peptidase, and β1 and β2 domains from the AF3 model, equivalent to ThomasA$^{31-878}$, fit in the head of this envelope, while the remaining β domains align along its longitudinal axis (Fig. 2G,H). Notably, the orientation of the β3–β6 domains in the AF3 model of ThomasA$^{31-1167}$ differs from that observed in our SAXS envelope, suggesting that their relative positioning in solution deviates from the predicted AF3 structure (Appendix Fig. S3).

## ThomasA catalytic domain specifically recognizes the Fab and the HR of IgA1 and IgA2

ThomasA has been shown to exhibit peptidase specificity for IgA1 and IgA2 allotype A2m(1), but not for IgA2 allotype A2m(2), IgG,

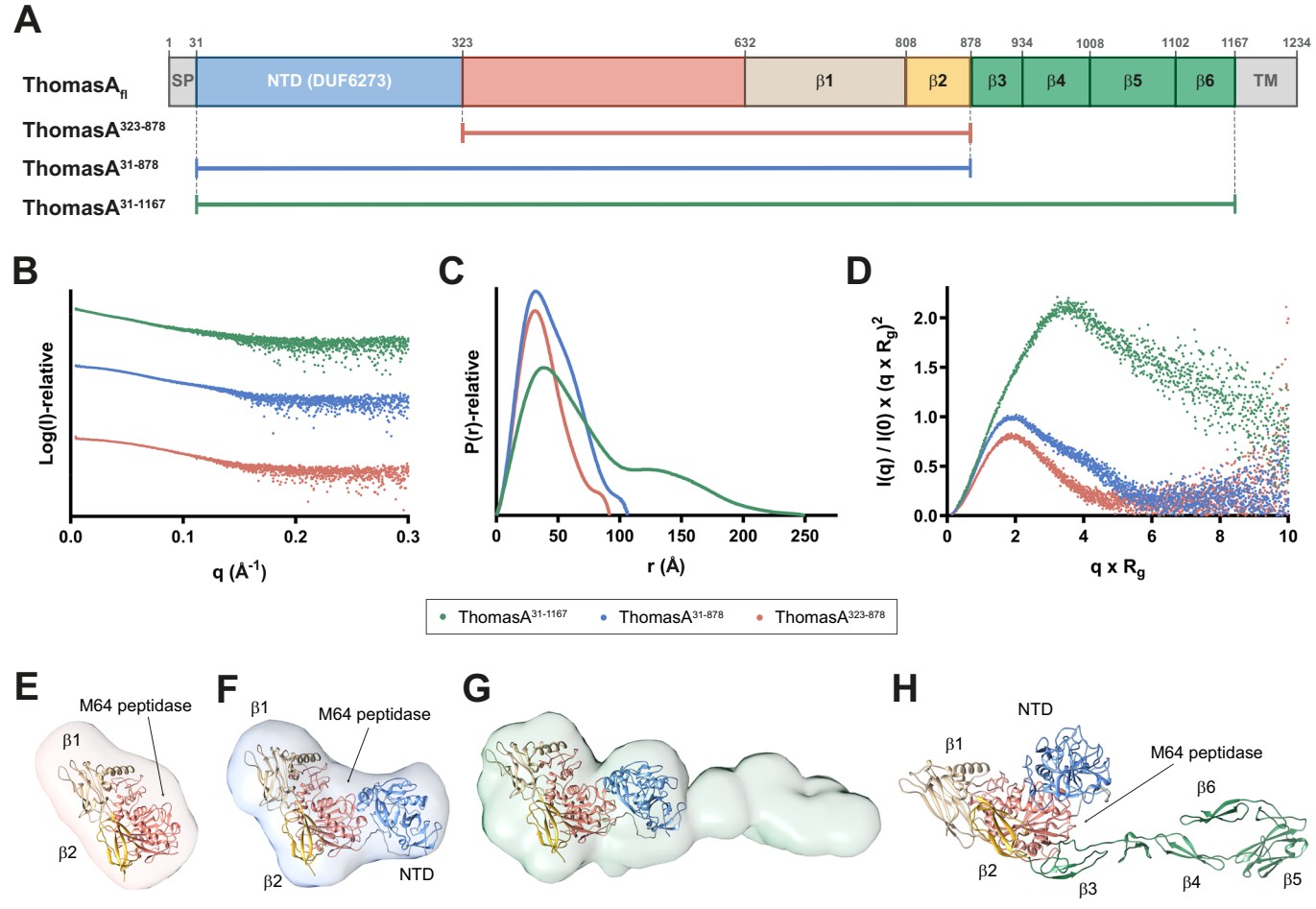

**Figure 2. Overall structure and domain organization of ThomasA.**

(A) Schematic representation of ThomasA full-length sequence and the ThomasA$^{323-878}$, ThomasA$^{31-878}$ and ThomasA$^{31-1167}$ constructs. (B) SAXS scattering curve of ThomasA$^{31-1167}$, ThomasA$^{31-878}$ and ThomasA$^{323-878}$. (C) P(r) function distributions of ThomasA$^{31-1167}$, ThomasA$^{31-878}$ and ThomasA$^{323-878}$. (D) Normalized Kratky plot of ThomasA$^{31-1167}$, ThomasA$^{31-878}$ and ThomasA$^{323-878}$. (E–G) Ab initio SAXS envelope of ThomasA$^{323-878}$ (E), ThomasA$^{31-878}$ (F) and ThomasA$^{31-1167}$ (G) using GASBOR. AF3 models of ThomasA$^{323-878}$ (E) and ThomasA$^{31-878}$ are superimposed (F, G). (H) AF3 model of ThomasA$^{31-1167}$ showing the NTD (blue), M64 peptidase (salmon), β1 (beige), β2 (gold) and β3–β6 (green) domains. Source data are available online for this figure.

IgD, IgE and IgM subclasses, or other proteins (Kosowska et al, 2002). The enzyme releases an intact Fc and two single Fab regions (Fig. 1A). To further investigate the role of ThomasA domains in IgA1 recognition, we analyzed the peptidase activity of ThomasA$^{31-1167}$, ThomasA$^{31-878}$ and ThomasA$^{323-878}$ against IgA1 over time. We incubated ThomasA constructs with full-length IgA1 (IgA1$_{fl}$) and monitored changes in intact IgA1$_{fl}$ and the released Fab region mass peaks by LC-MS at ~6-min intervals (Fig. 3A–C; Appendix Fig. S5A–D). We observed a similar activity among the three constructs, which suggests that IgA1 recognition is mediated exclusively by the M64 peptidase domain in combination with the β1 and β2 domains (Fig. 3A–C).

Having identified the domains responsible for the hydrolytic activity of ThomasA, we next examined the specific IgA domains required for enzyme recognition and cleavage. To this end, we designed a two-domain fusion protein consisting of the Maltose-Binding Protein (MBP) domain and the artificially modified Green Fluorescent Protein (EGFP) connected by a long linker with a TEV site and the IgA1 HR, MBP-TEV-IgA1$_{HR}$-EGFP (Appendix Fig.

S6A,F). We evaluated the peptidase activity of ThomasA$^{31-1167}$, ThomasA$^{31-878}$ and ThomasA$^{323-878}$ against this construct and monitored the reaction by SDS-PAGE (Appendix Fig. S6B). None of the ThomasA constructs were able to hydrolyze the IgA1 HR within MBP-TEV-IgA1$_{HR}$-EGFP over time, suggesting that the enzyme requires the Fab and/or Fc regions of IgA1 for recognition and interaction with the substrate (Appendix Fig. S6B).

To further dissect the enzyme requirements, we designed two IgA1 chimeras: one containing the Fab region and IgA1 HR fused to the Fc region of IgG1 (IgA1$_{Fab-HR}$-IgG1$_{Fc}$) and another with the Fab region of IgG1 fused to the HR and Fc region of IgA1 (IgG1$_{Fab}$-IgA1$_{HR-Fc}$, Fig. 3D). The first chimera failed to express, but we were able to evaluate the proteolytic activity of ThomasA$^{31-1167}$, ThomasA$^{31-878}$, and ThomasA$^{323-878}$ against IgG1$_{Fab}$-IgA1$_{HR-Fc}$ by LC-MS (Fig. 3E; Appendix Figs. S5H and S6C–E,G). None of the enzyme constructs exhibited activity against this chimera, suggesting that the presence of the IgA1 HR and Fc region alone is insufficient for ThomasA to catalyze hydrolysis. To assess the role of the Fab region in the activity of the enzyme, we purified two

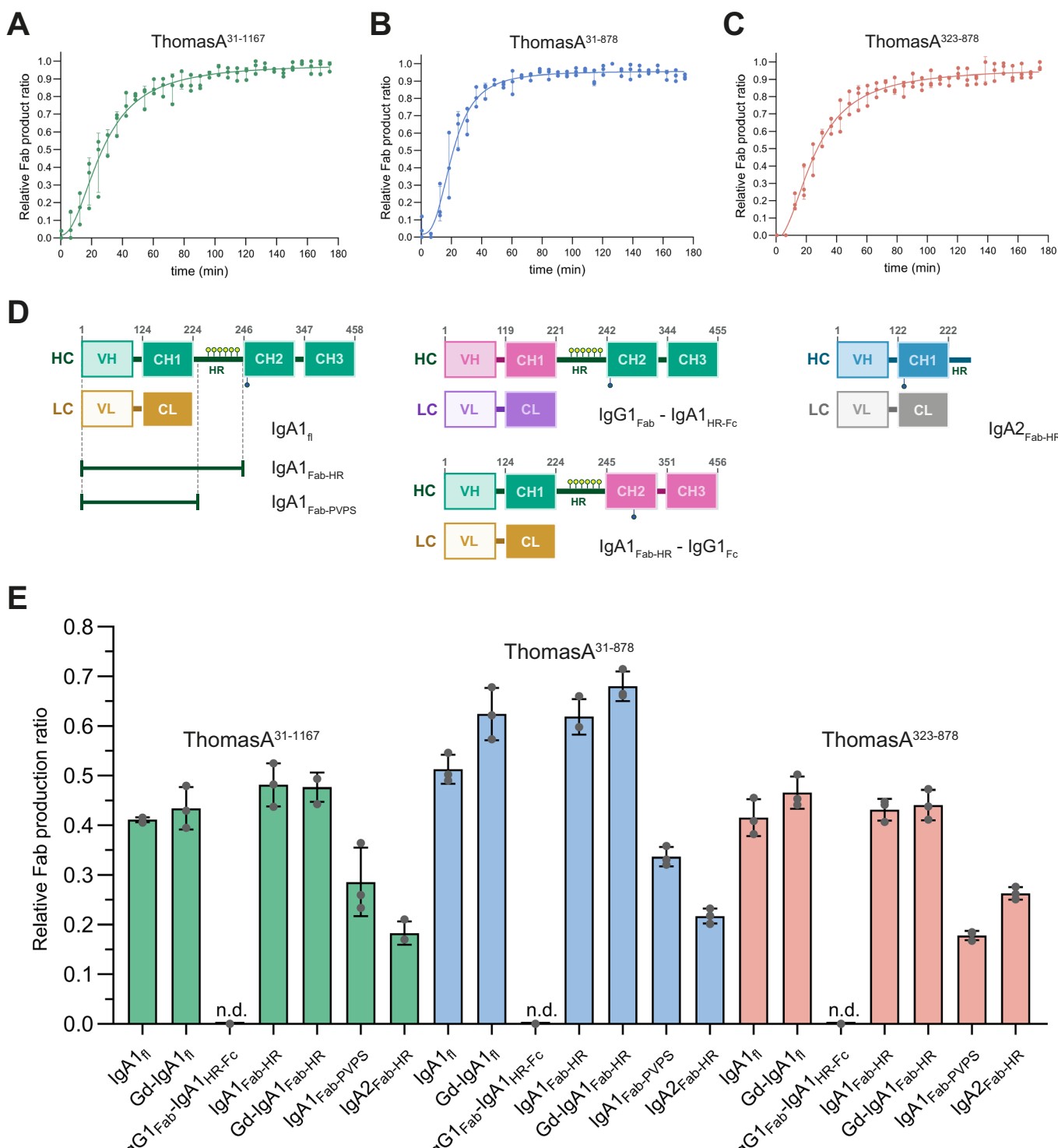

**Figure 3. Hydrolytic activity of ThomasA against IgA substrates by LC-MS.**

(A–C) Time-course analysis of Fab product generation from IgA1fl substrate by ThomasA[31-1167] (A), ThomasA[31-878] (B) and ThomasA[323-878] (C) over 3 h. (D) Schematic representations of IgA constructs designed and produced to study ThomasA specificity. IgA1 *O*-glycosylation is represented in yellow circles. (E) Single point measurements at 45 min of hydrolytic activity of ThomasA[31-1167] (green), ThomasA[31-878] (blue) and ThomasA[323-878] (salmon) against IgA constructs. n.d. not detected. Results are expressed as mean ± SD from three independent biological replicates (*n* = 3). Source data are available online for this figure.

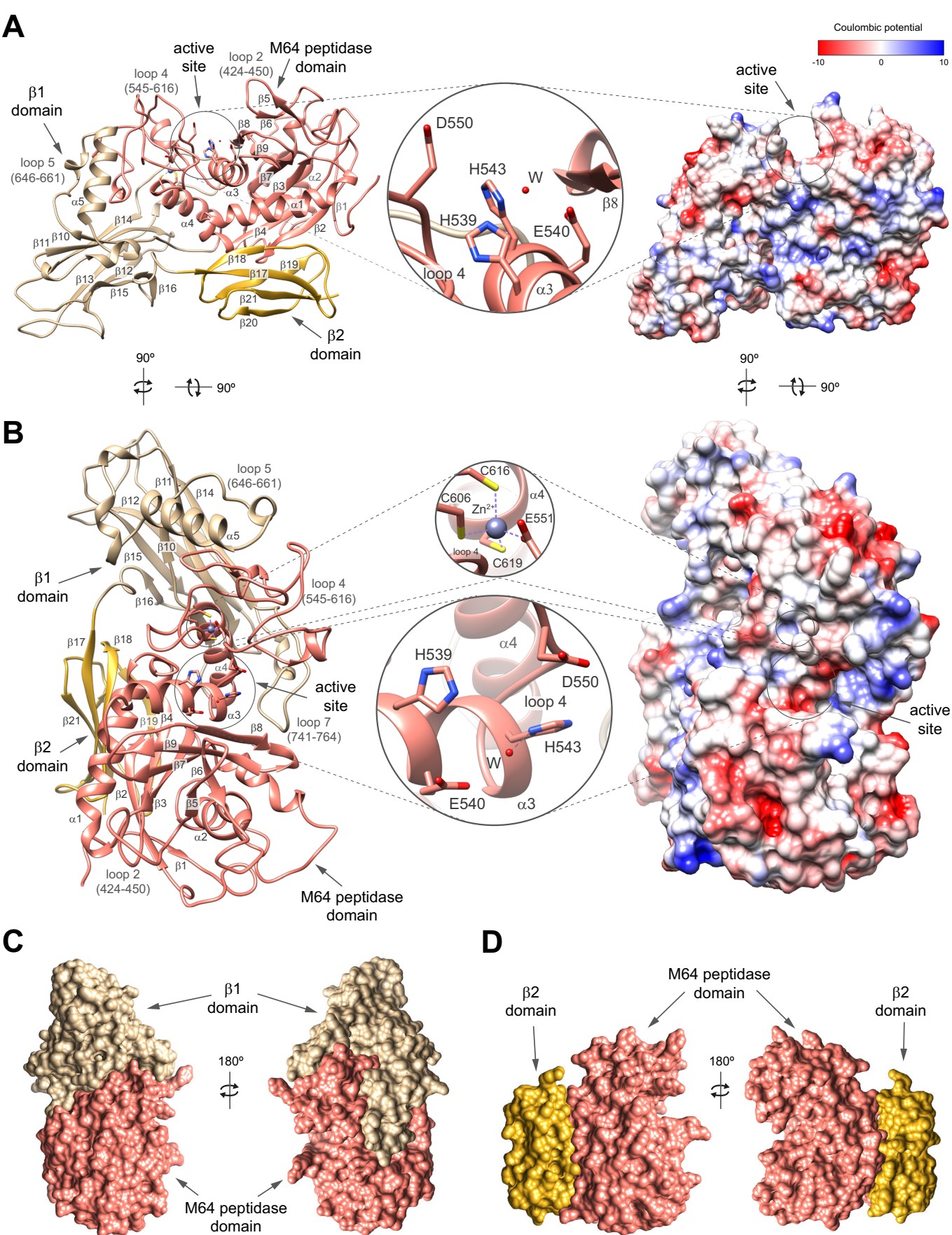

**Figure 4. X-ray crystal structure of ThomasA$^{323-878}$.**

(A, B) Two views of the overall fold and secondary structure organization of ThomasA$^{323-878}$ in cartoon (left) and electrostatic surface representations (right), including the M64 peptidase domain (salmon), β1 (beige), and β2 (gold) domains. Close-up view of the active site (A, B) and secondary binding site for Zn$^{2+}$ (B) of ThomasA$^{323-878}$ shown as cartoon/stick representation. (C, D) Two surface views of the shared interfaces between M64 peptidase and β1 domains (C) and β2 domains (D) of ThomasA$^{323-878}$.

IgA1 Fab constructs with different HR lengths. IgA1$_{Fab-HR}$ contained the complete HR sequence (221–240) while IgA1$_{Fab-PVPS}$ contained only the residues VPS following the cleavage site of the enzyme (Fig. 3D). We also purified an IgA2 Fab fragment containing the sequence of the HR (IgA2$_{Fab-HR}$, Fig. 3D). We then tested the hydrolytic activity of ThomasA constructs against these substrates after 45 min of reaction using LC-MS (Fig. 3E; Appendix Fig. 5I,J,M–P). All three ThomasA constructs were able to hydrolyze the Fab regions of both IgA1 and IgA2; however, they exhibited reduced activity against the IgA1 construct with a shorter HR, IgA1$_{Fab-PVPS}$, and IgA2$_{Fab-HR}$. These findings suggest that ThomasA preferentially recognizes the Fab region of IgA and that the HR length and flexibility may influence antibody recognition by the enzyme.

The HR of IgA1 is heavily glycosylated, with three to six O-glycans (Fig. 1A,B). To assess the impact of O-glycosylation on ThomasA activity, we enzymatically removed Gal and Neu5Ac residues from both O-glycans and N-glycans of IgA1$_{fl}$ and IgA1$_{Fab-HR}$ using GalactEXO and SialEXO to obtain Gd-IgA1$_{fl}$ and Gd-IgA1$_{Fab-HR}$ (Appendix Fig. S5E–G,K,L). We then evaluated the activity of ThomasA against these partially deglycosylated proteins using LC-MS (Fig. 3E). All three ThomasA constructs exhibited similar activity to that observed with fully glycosylated IgA1 samples, suggesting that enzyme recognition and hydrolysis are not significantly affected by the carbohydrates that form the O-glycans in the HR (Fig. 3E).

## Structural insights into the ThomasA catalytic domain

We determined the X-ray crystal structure of ThomasA$^{323-878}$ in apo form at 2 Å resolution (PDB code 9QDH, Fig. 4; Appendix Table S3, Appendix Figs. S7 and S8). Although the crystals were grown in the presence of the decapeptide VPCPVPSTPP, which corresponds to a segment of the IgA1 HR, no density for the peptide was observed in the active site of the structure. ThomasA$^{323-878}$ comprises three distinct domains arranged from the N- to the C-terminus: an M64 peptidase domain (residues 323–632) and two β-sandwich domains (β1 domain: residues 633–807; β2 domain: residues 808–878) (Fig. 4). The M64 peptidase domain adopts a seven-stranded β−sheet topology (β1–β2–β4–β3–β7–β9–β8), with β2 and β8 in an antiparallel arrangement. This core is flanked by three α-helices, α1, α2 and α3 and the connecting loop β4–α2 (loop 2, residues 424–450), which contains a β-hairpin formed by β5 and β6. The active site, which extends longitudinally across the enzyme, is a large, solvent-accessible cavity flanked by β8, α3, loop 2, connecting loop α3–α4 (loop 4, residues 545–616) from the M64 domain and by a small segment of connecting loop β13–β14 (loop 7, residues 741–764) from the β1 domain. However, we did not observe electron density for the predicted catalytic Zn$^{2+}$ that should be coordinated with two histidine residues (H539 and H543) from α3, along with D550 from a long connecting loop 4 (Fig. 4A,B). These residues are conserved in the metzincin metallopeptidase motif **HEXXHXXXGXXD**, which is critical for ThomasA

hydrolytic activity (Kosowska et al, 2002). A secondary Zn$^{2+}$ binding site is located within the M64 peptidase domain, where it adopts a tetragonal coordination with E551 of loop 4 and three cysteine residues (C606, C616, C619) from loop 4 and α4. This secondary site is less solvent-exposed, suggesting a structural rather than a catalytic role (Laitaoja et al, 2013) (Fig. 4B).

The M64 peptidase domain comprises over one-third of the total protein volume. The β1 and β2 domains encapsulate the M64 peptidase domain on the side opposite the active site, burying 4714 and 2026 Å$^2$ of the interface, respectively (Fig. 4C,D) according to PISA analysis (Krissinel and Henrick, 2007). The β1 domain consists of two opposing antiparallel β−sheets (β10–β11–β14 and β13–β12–β15–β16). Its connecting loop β10-α5 (loop 5, residues 646–661), along with α5 and loop 7, interacts with loop 4 of the M64 peptidase domain, suggesting a potential role in domain stabilization (Fig. 4A–C). The β2 domain is smaller and comprises two opposing β-sheets: one composed of three β-strands (β17–β21–β20) and the other of two β-strands (β18–β19), both of which engage with the core of the M64 peptidase domain (Fig. 4A,B,D). The overall molecular surface of ThomasA$^{323-878}$ is predominantly polar, with no pronounced hydrophobic or highly charged regions, except for the slightly negatively charged Zn$^{2+}$ catalytic pocket (Fig. 4B). In addition, using CRYSOL (Svergun et al, 1995), we observed a moderate fit of the SAXS data to the solution scattering profile calculated from the crystal structure of ThomasA$^{323-878}$, suggesting that this structure is largely conserved in solution (Appendix Fig. S9).

## BF3526 from *B. fragilis* exhibits distinct substrate specificity compared to ThomasA

To further investigate the substrate specificity of M64 peptidases, we purified BF3526 from *Bacteroides fragilis* NCTC 9343 (UniProt code Q5L9L3, residues 19–426), the closest reported structural homolog of the M64 peptidase domain of ThomasA available in the PDB database. Our SSN analysis classified BF3526 within CL-I, the most populated cluster of the M64 family. We tested the peptidase activity of BF3526 against human IgA1 from myeloma (hIgA1), anti-hEGFR IgA2 (hIgA2) and IgG1 (Trastuzumab) (Fig. 5A). SDS-PAGE analysis revealed that, unlike ThomasA, which specifically cleaves hIgA1 and hIgA2—yielding a single-chain Fc (scFc) and the heavy-chain fragment of the Fab (Fd)—BF3526 failed to hydrolyze any of the tested substrates (Fig. 5A,B).

Given the distinct substrate specificity of BF3526 compared to ThomasA, we further evaluated its ability to hydrolyze hIgA1 and hIgA2 pretreated with other IgAPs. hIgA1 was treated with the M26 family peptidase IgASAP Sub1 and the M64 family peptidase IgASAP Sub1 + 2, while hIgA2 was treated with IgASAP Sub1 + 2 (Fig. 5; Appendix Fig. S10). These enzymes cleave IgAs, exposing the HR and potentially facilitating further digestion by BF3526. After overnight incubation of BF3526 at 37 °C with the resulting IgA fragments Fab and Fc, LC-MS analysis revealed additional hydrolysis of the HR in the

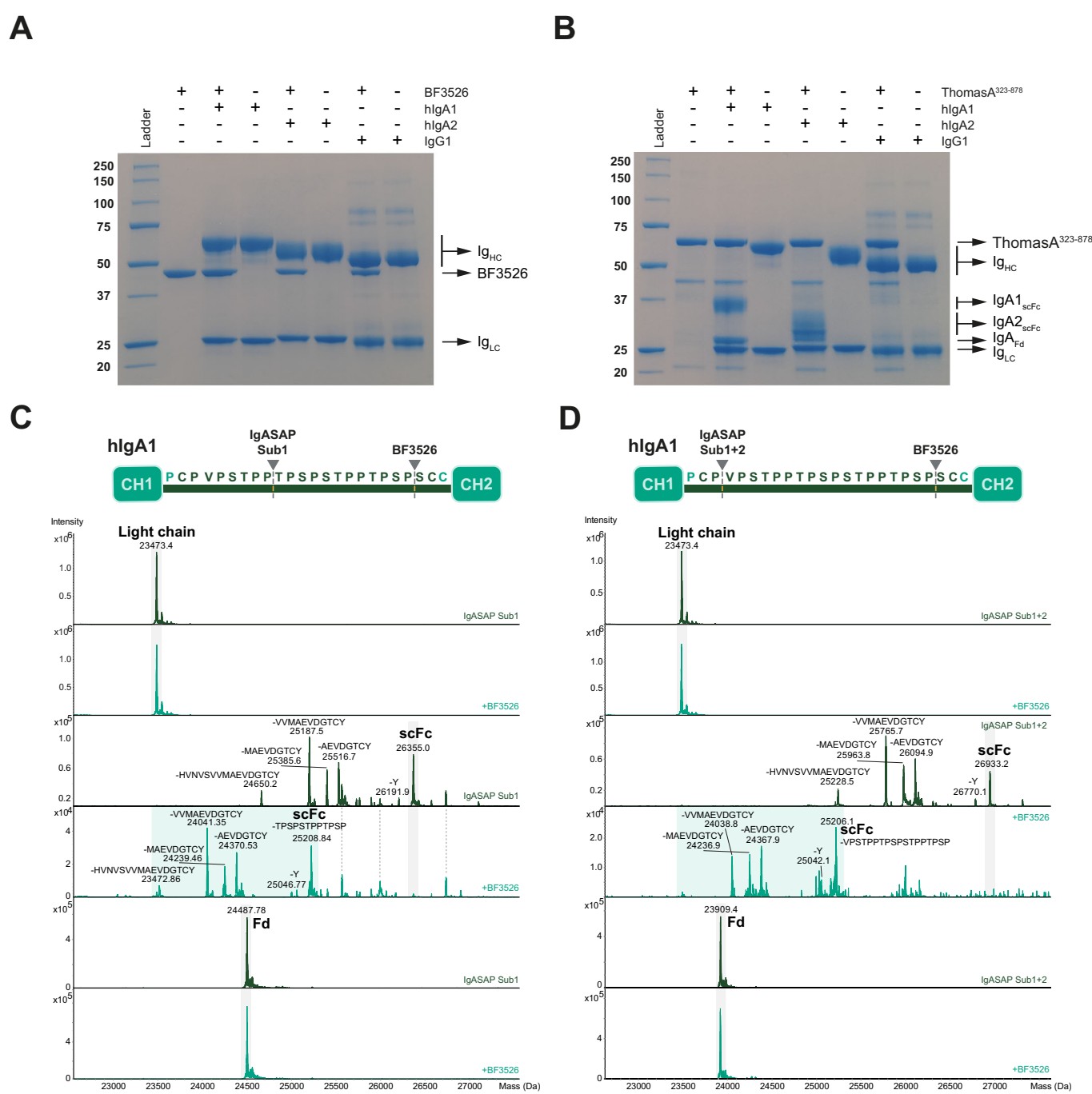

**Figure 5. Hydrolytic activity of BF3526.**

(A, B) SDS-PAGE analysis of the hydrolytic activity of BF3526 (A) and ThomasA[323-878] (B) against hIgA1, hIgA2 and IgG1. (C, D) LC-MS analysis of hydrolytic activity of IgASAP Sub1 alone (dark green) or with BF3526 (light green) (C) and IgASAP Sub1 + 2 alone (dark green) or with BF3526 (light green) (D) against LC, scFc and Fd fragments of hIgA1. To facilitate data analysis, IgA1 was deglycosylated using PNGase F and *O*-glycan-related exo-glycosidases prior to protease digestion. All enzymatic activity measurements were performed in triplicate. Source data are available online for this figure.

Fc fragments of both antibodies (Fig. 5C,D; Appendix Fig. S10), whereas the Fab fragment remained intact. Specifically, BF3526 predominantly cleaved hIgA1 and hIgA2 after a proline residue in the Fc fragment, with minor cleavage occurring after a cysteine residue of hIgA2 (Appendix Fig. S10). To further characterize BF3526's cleavage preferences, we performed peptide mapping analysis of trypsin-digested hIgA2 and Lys-C-digested IgG1 (Appendix Fig. S11). BF3526 cleaved primarily near the N-terminus and in proximity to proline, though not exclusively. Additional cleavage sites were observed after serine, glycine, and threonine.

To extend our analysis, we tested BF3526 against a synthetic decapeptide (VPCPVPSTPP) which corresponds to a segment of the

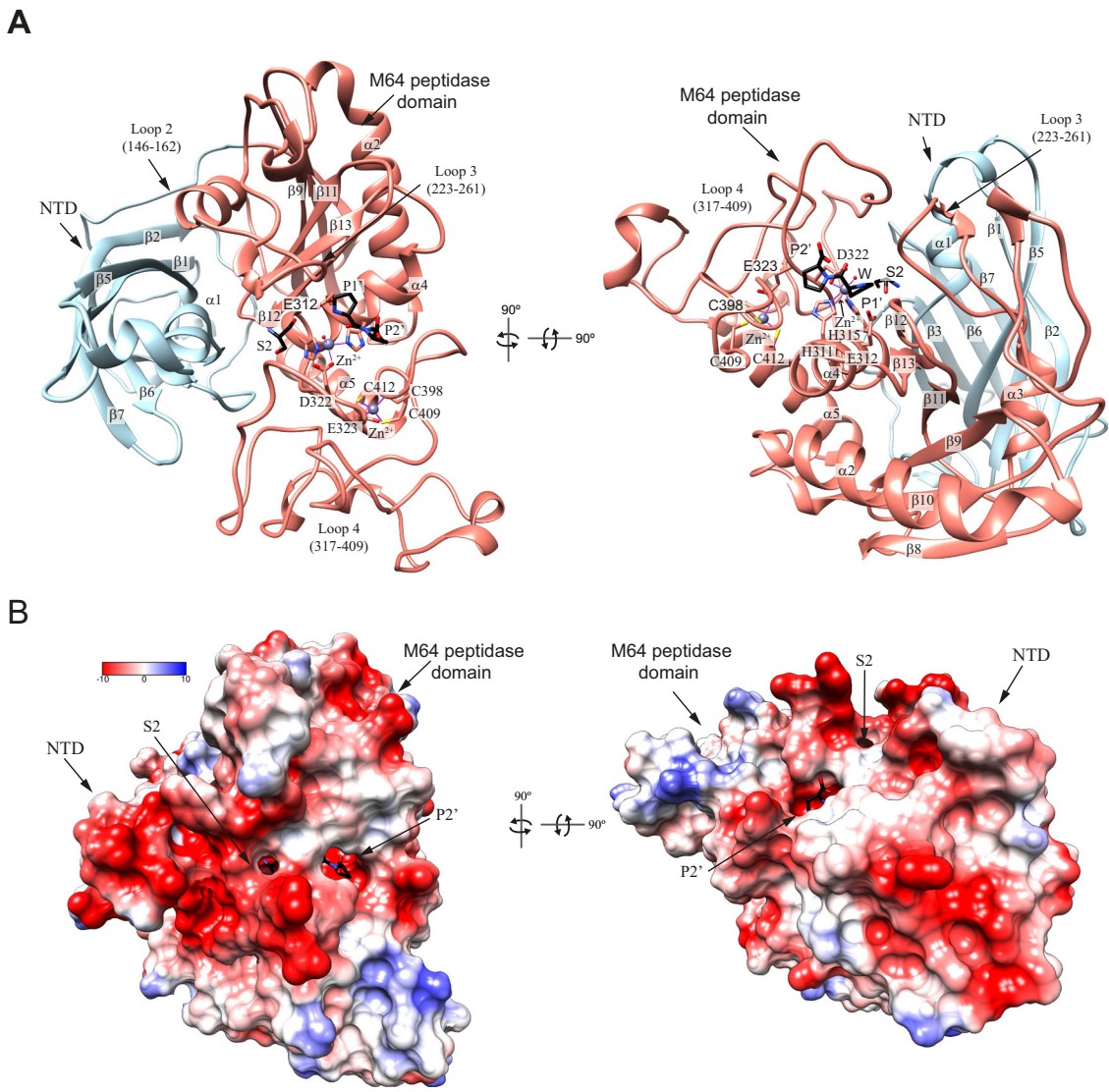

**Figure 6. X-ray crystal structure of BF3526-PP.**

(A) Two views of the overall fold and secondary structure organization of BF3526-PP in cartoon representation, including the β-sandwich (blue) and M64 peptidase domain (salmon). (B) Two electrostatic surface representations of BF3526-PP showing the active site. The polar character of the enzyme is shown in three colors: negatively charged patches in red (−10 kcal mol−1), hydrophobic zones in white (0 kcal mol−1), and positively charged residues in blue (10 kcal mol−1).

IgA1 HR, and analyzed the outcome using LC-MS (Fig. EV1). At an enzyme-to-substrate ratio of 1:100, BF3526 hydrolyzed the peptide into species undetectable by this method within 5 min. When the ratio was reduced to 1:10,000, two new peaks corresponding to CPVPSTPP and VPSTPP appeared after 1 h of incubation, suggesting that VP-CPVPSTPP and VPCP-VPSTPP were the preferred cleavage sites of BF3526. These peptides were further degraded into smaller fragments that were undetectable by LC-MS. Interestingly, ThomasA[323-878] was not able to hydrolyze the peptide VPCPVPSTPP (Fig. EV1).

## The X-ray crystal structure of BF3526 from *B. fragilis* in complex with substrate and product peptides

We performed co-crystallization experiments with the full-length wild-type BF3526 in the presence of the decapeptide VPCPVPSTPP

(PDB code 9QDI, Figs. 6 and 7; Appendix Figs. S7, S12, 13 and Appendix Table S3). BF3526 crystallized in the triclinic space group P1 with eight molecules in the asymmetric unit (ASU) and diffracted to a maximum resolution of 1.9 Å (Appendix Table S3). We could model the tetrapeptide substrate STPP in two molecules of the ASU (BF3526-STPP), and the dipeptide product PP in two others (BF3526-PP), while the peptide did not occupy the active site in the remaining four molecules (BF3526-unliganded). These findings suggest that the enzyme hydrolyzed the decapeptide, trapping the STPP peptide in the active site during crystallization, where it continued to undergo enzymatic cleavage. No major conformational changes are observed across the three enzyme states, except for connecting loop β10-α3 (loop 3, residues 223–261), which shifts closer to the active site in the peptide-bound structures, and connecting loop α4–α5 (loop 4, residues

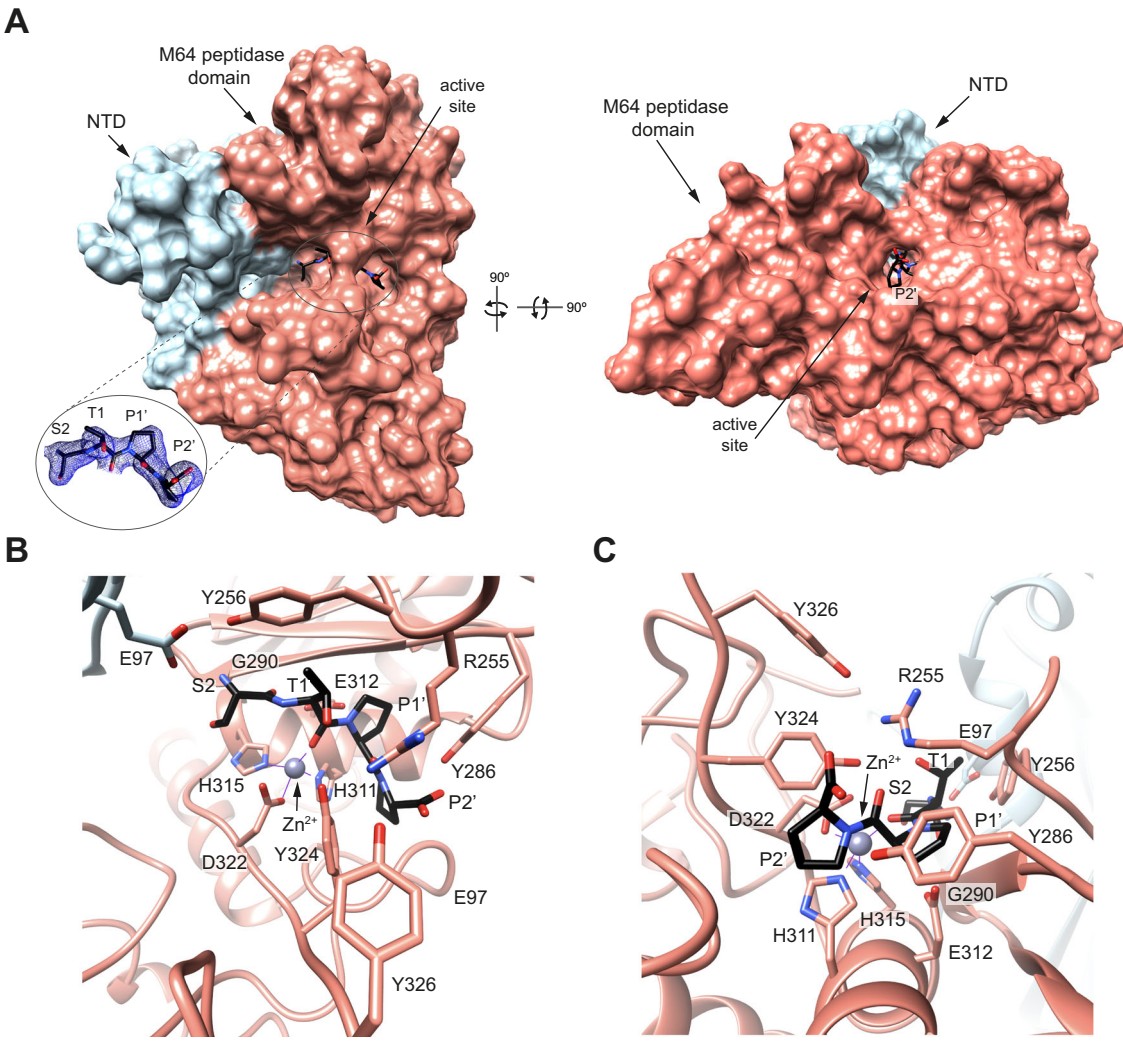

**Figure 7. X-ray crystal structure of BF3526-STPP.**

(A) Two views of the overall fold of BF3526-STPP in surface representation, highlighting the NTD (blue) and the M64 peptidase domain (salmon). Close-up view of the electron density map of the S2T1P1'P2' substrate (left), shown at 1.0 σ r.m.s.d. (B, C) Two views of the BF3526-STPP structure in cartoon representation, showing the location of S2T1P1'P2' substrate within the active site and the key interacting residues.

317–409), which is only structured in chain C of the BF3526-PP complex, suggesting flexibility above the active site (Fig. EV2; Appendix Table S4). This chain has therefore been used for the overall structural descriptions of the protein. BF3526 comprises two domains from the N- to the C-terminus: N-terminal β-sandwich domain (NTD, residues 21–145) followed by a catalytic M64 peptidase domain (residues 163-426). These domains are connected by a short linker (loop 2, residues 146–162) (Fig. 6). The NTD consists of two opposing antiparallel β−sheets (β5−β1−β2 and β4−β3−β6−β7), with a short α-helix, α1, between β4 and β5. The catalytic M64 peptidase domain shares a similar topology with ThomasA[323-878] structure, featuring a six-stranded β-sheet (β8−β10−β9−β11−β13−β12, with β8 and β13 antiparallel) instead of seven, flanked by three α-helices (α2, α3, α4) and the loop 3 (Fig. 6A). In the BF3526-PP, BF3526-STPP and BF3526-unliganded molecules, the catalytic Zn[2+] ion is buried in the active site flanked by α4, β12, loop 3, loop 4 and NTD (Figs. 6 and EV2). In this

BF3526-PP molecule, loops 3 and 4 close the active site through interactions between R255 of loop 3 and the side chains of Y324 and D329 of loop 4, as well as Y256 of loop 3 and M331 of loop 4 (Figs. 6 and EV3). Similar to ThomasA[323-878] structure, a secondary Zn[2+] binding site is located within the M64 peptidase domain, where it adopts tetragonal coordination with E323, C398 and C409 of loop 4 and C412 of α4 (Fig. 6A). A structural homology search using the DALI server (Holm et al, 2023) revealed only one protein with high structural homology to BF3526, BACOVA_00663 from *B. ovatus* ATCC 8483 (PDB code 3P1V; Z-score of 66.2; r.m.s.d. value of 0.9 Å for 405 aligned residues; 73% identity; Appendix text, Appendix Fig. S14).

The STPP substrate and PP product were unambiguously modeled in the active site of the enzyme, occupying equivalent positions within the narrow cavity flanked by α4, β12, and loops 3, 4 and NTD (Figs. 6A,B, 7A, EV2 and 3). By convention, substrate side chains upstream of the scissile bond are designated as P1, P2,

P3, etc., while those downstream are labeled P1', P2', P3', etc. Following this nomenclature, we assigned the tetrapeptide STPP as S2T1P1'P2'. In the BF3526-STPP molecules, we observe an extensive network of interactions between the peptide and the protein (Fig. 7). Specifically, the S2 main chain forms hydrogen bonds with the side chains of Y256 of loop 3 and E97 in α1 from the NTD, as well as with the main chain of G290. Meanwhile, T1 side chain interacts with Y256 and R255, while its scissile carbonyl forms hydrogen bonds with Y324. Notably, the scissile carbonyl oxygen of T1 also coordinates with the catalytic $Zn^{2+}$ ion, which is further coordinated by H311 and H315 from α4, and D322 of loop 4 in a tetrahedral geometry (Fig. 7B,C). The main chain of P1' also makes hydrogen bonds with the side chain of Y286 and its side chain makes additional interactions with R255 and Y286. In addition, the residue P2' makes hydrophobic interactions with the side chain of H311 and Y324 and polar interactions with Y286.

In the BF3526-PP molecules, the dipeptide PP adopts an identical conformation to its equivalents in the BF3526-STPP molecules, engaging in similar interactions with the protein (Fig. EV3). Additionally, we identified a serine residue occupying the same position as in the substrate molecules, but no clear electron density was observed for the remaining threonine (Fig. EV3 and Appendix Fig. S13). The $Zn^{2+}$ ion maintains a tetrahedral geometry, coordinated with H311, H315, D322, and a water molecule (Figs. 6 and EV3). Conversely, in these molecules, the N-terminal nitrogen atom of P1' forms strong hydrogen bonds with the water molecule coordinated to the zinc atom and with one oxygen atom of the base/acid E312 (Fig. EV3B). In both BF3526-PP and BF3526-STPP molecules, loop 3 shifts 2.9 Å toward the active site compared to the BF3526-unliganded molecules (Fig. EV2). This movement may allow Y256 in loop 3 to engage in hydrophobic interactions with T1 or a proline at this position, the latter being the preferred residue for enzymatic hydrolysis.

Our X-ray crystal structure of BF3526, solved in both its unliganded form and in complex with substrate and product peptides, supports a single-displacement catalytic mechanism, consistent with those established for metallopeptidases (Cerdà-Costa and Gomis-Rüth, 2014; Dudev and Lim, 2000; Auld, 2013; Vallee and Auld, 1990; Bertini et al, 2006; Matthews and Matthews, 2002; Pelmenschikov and Siegbahn, 2002) (Appendix Fig. S15). In the holoenzyme, the water molecule is coordinated to the metal atom, and it is polarized by the base/acid glutamate which enhances its nucleophilicity. Once the peptide substrate is bound in the active cleft, the scissile carbonyl oxygen is polarized by the catalytic $Zn^{2+}$ atom, thus the water molecule is able to attack the carbonyl oxygen of the peptide and simultaneously transfer a proton to the general base glutamate. This reaction produces a *gem*-diolate tetrahedral intermediate that is stabilized by the $Zn^{2+}$ atom and neighboring residues. In the BF3526-STPP molecules, the scissile carbonyl oxygen of the bound STPP peptide coordinates with $Zn^{2+}$, suggesting a pre-catalytic state prior to the formation of the *gem*-diolate via reaction with a water molecule, for which no clear electron density was observed. In the fully refined X-ray BF3526-STPP structure, additional electron density was observed in the active site, potentially indicating a mixture of substrate conformations or other reaction components (Appendix Fig. S15C,D). However, the omit map suggests that the predominant species is the substrate in these molecules (Appendix Fig. S15D). Cleavage of the scissile bond results in a double-product complex, with proton transfer to the newly formed α-amino terminus. Finally, both products are released, likely with the nonprimed product dissociating first, as in the BF3526-PP molecules of our structure.

## The NTD regulates substrate accessibility of BF3526

The X-ray crystal structure of BF3526 suggests that the NTD may impose structural constraints on the entry of the protein's N-terminus into the enzyme's active site (Figs. 6, 7 and EV2). To directly assess the activity of the M64 peptidase domain, we attempted to express BF3526 without the NTD (BF3526[161-426]) and test its activity against IgA1 and hIgA2. However, the truncated domain failed to express on its own. Given that BF3526 originates from the commensal *B. fragilis*, which colonizes the human intestinal tract and interacts with both human and bacterial peptidases, we examined its stability in the presence of human digestive enzymes. Specifically, treatment with chymotrypsin resulted in the digestion of BF3526 (BF3526$_D$) into three fragments (Appendix Fig. S16). LC-MS analysis of the BF3526$_D$ SDS-PAGE bands identified one fragment as the NTD (BF3526$_{NTD}$), while the other two corresponded to the C-terminal portion (BF3526$_{CT1}$ and BF3526$_{CT2}$) (Appendix Fig. S16D). To determine whether the proteolytic fragments retained enzymatic activity, we tested them against IgA1$_{fl}$, hIgA2 and albumin (Appendix Fig. S16B). SDS-PAGE analysis revealed that the BF3526$_D$ fragments hydrolyzed IgA1$_{fl}$ and hIgA2 into nonspecific smaller fragments. These results suggest that the NTD acts as a barrier, restricting access to proteins with a short or highly flexible N-terminus.

## Structural basis for IgA1 recognition by M64 metallopeptidases

Structural comparison of ThomasA[323-878] and BF3526-STPP crystal structures revealed a conserved overall architecture of the M64 peptidase domain (Fig. 8A). However, the loops and α-helices connecting the β-strands of the β-sheet core differ significantly between the two enzymes. These structural differences may contribute to their distinct substrate specificities, despite both enzymes being classified within the same peptidase family. Additionally, based on our BF3526-STPP structure, we modeled the tetrapeptide C2P1V1'P2' in the active site, which is part of the HR cleavage site for ThomasA and can also be hydrolyzed by BF3526 (CP-VP). Superposition of this structure and that of ThomasA[323-878] revealed that the peptide could be accommodated in the active site, but its interactions with the enzyme differ from those observed in BF3526 (Fig. 8B). Specifically, the main chain of C2 interacts with the main chain of A513 from β12 in ThomasA, while its side chain contacts H543, which adopts a different conformation compared to BF3526 (H315) in the absence of $Zn^{2+}$. Interestingly, P1 engages the base/acid residue E540 of ThomasA, equivalent to E312 in BF3526 and interacts with the main chain of G511 and G512 in β8 of ThomasA. In contrast, Y552 in ThomasA, equivalent to Y324 in BF3526, adopts a distinct conformation. Moreover, Y436 in ThomasA, which could correspond to Y256 in BF3526 (a key residue interacting with T1 in the BF3526-STPP structure), is positioned too far from the active site to form comparable interactions. The side chain of V1' interacts with the main chain of G511, while P2' contacts R532 and H539 in ThomasA, equivalent to N304 and H311 in BF3526. However, the

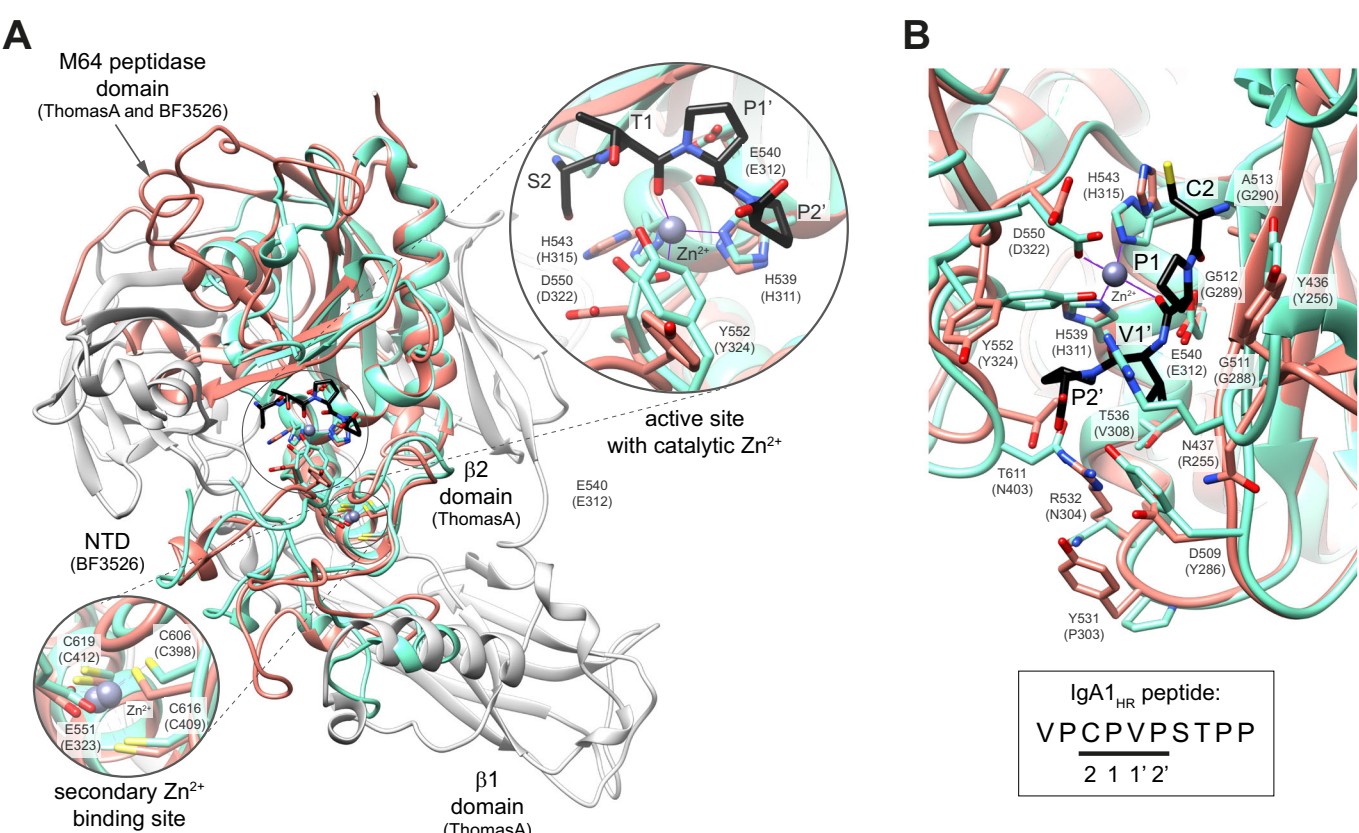

**Figure 8. Structural comparison of ThomasA³²³⁻⁸⁷⁸ and BF3526-STPP structures.**

(**A**) Structural superposition of M64 peptidase domains of ThomasA³²³⁻⁸⁷⁸ (salmon, PDB code 9QDH) and BF3526-STPP (aquamarine, PDB code 9QDI), with a close view of the secondary Zn²⁺ binding and active sites. Accessory domains of both proteins are colored in light grey. BF3526 equivalent residues are in parentheses. (**B**) Structural superposition of BF3526-STPP active site with CPVP fragment manually docked in ThomasA³²³⁻⁸⁷⁸ active site. Zn²⁺ catalytic ion is placed according to the BF3526 structure.

overall number of peptide–protein interactions in ThomasA is significantly lower than in the BF3526-STPP complex (Fig. 8B).

To explore the molecular mechanism of IgA1 recognition by ThomasA, we attempted to purify the native complex of inactive ThomasA³²³⁻⁸⁷⁸-H539A/E540A with the IgA1$_{Fab-HR}$ (ThomasA-IgA1), by mixing a 1:5 ratio of the enzyme and substrate. However, complex formation was not observed (Appendix Fig. S17A). Additionally, cross-linking experiments using various reagents did not yield a detectable complex in SDS-PAGE analysis (Appendix Fig. S17B). Despite extensive efforts, we were unable to isolate a stable native or cross-linked ThomasA-IgA1 complex. To investigate the molecular basis of ThomasA–IgA1 interaction and based on our experimental data, we generated two AF3 models using ThomasA³²³⁻⁸⁷⁸ in combination with either the CH1 domain (IgA1$_{CH1}$) or the CH1-CL domains (IgA1$_{CH1-CL}$) from our IgA1$_{Fab-PVPS}$ construct (Fig. EV4). Both models predicted by AF3 exhibited a consistent overall structure with high accuracy in the interaction interface (Fig. EV4). In the ThomasA³²³⁻⁸⁷⁸-IgA1$_{CH1}$ AF3 model, the CH1 was interacting exclusively with the M64 peptidase domain of ThomasA (Figs. 9A and EV4A,B), suggesting that the β1 and β2 domains do not contribute to antibody recognition. Surprisingly, the short HR (PVPS) of the IgA1$_{Fab-PVPS}$ that includes the active site of ThomasA is not in close proximity to the active site of the enzyme (~13 Å between the oxygen OE1 of

E540 distance and the carbon CA of V1' from the HR), suggesting that the flexible HR would require conformational changes to enter the active site of the enzyme (Figs. 9A and EV4A,B).

Based on this predicted model, we performed single alanine scanning mutagenesis on ThomasA³²³⁻⁸⁷⁸ and IgA1$_{fl}$ to identify key residues affecting enzymatic hydrolysis. We determined the relative percentage of hydrolyzed Fab and its mutants using ThomasA wild-type and selected mutants at 45 min of reaction by intact protein LC-MS analysis (Fig. 9B,C). Specifically, we introduced the following alanine mutations in ThomasA: W340 of loop 1 (residues 335-350); K435, Y436, N437, N444 and W450 of loop 2; H453 of α2; E486 and Y487 of loop 3 (residues 468-498); N516 and Y519 from β8; and K561 of loop 4, which flanks the active site (Fig. 9A,B). In the CH1 domain of the Fab region, we mutated K126, S131 and L132 from β-strand A; S161 from the CD loop; D181 from the DE loop; T194, Q195 and K200 from the EF loop; S201 and H205 from the β-strand F; N211 and P212 from the FG loop; and S213, D215 and C220 from β-strand G (Fig. 9A,C). The alanine mutagenesis analysis highlights the importance of H453 in ThomasA for hydrolytic activity, which in our AF3 model interacts exclusively with D215 of CH1 domain (Fig. 9A). Notably, mutating the H453 of ThomasA or the D215 of CH1 domain to alanine resulted in the loss of hydrolytic activity (Fig. 9B,C). Similarly, the K435A mutation in ThomasA also abolished the enzyme's

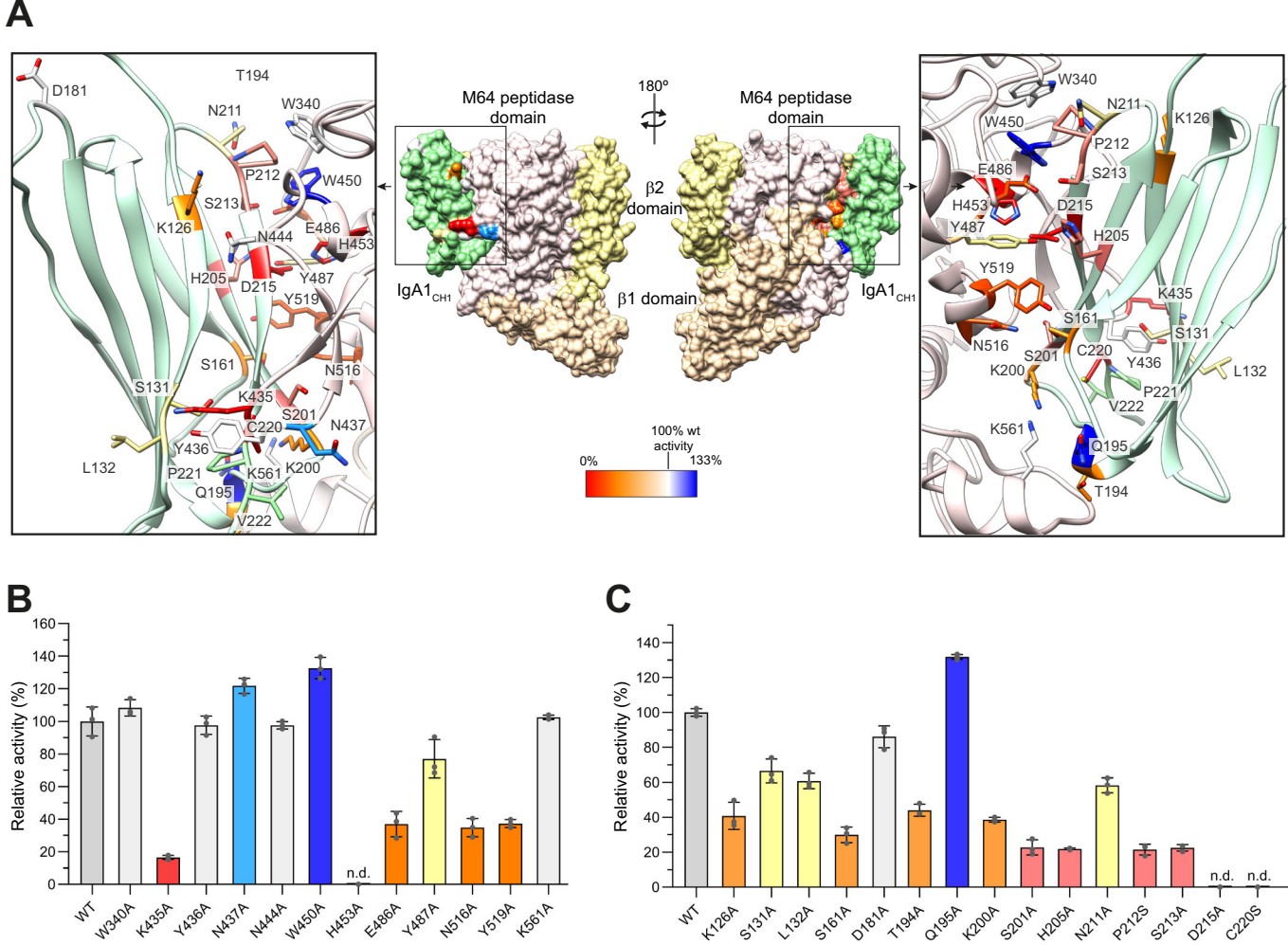

**Figure 9. Structural basis of ThomasA IgA1 specificity.**

(A) Two views of the AF3 model of ThomasA$^{323-878}$-IgA1$_{CH1}$ complex in cartoon and surface representation. Close-up view of the interaction interface depicted within the black rectangles. (B, C) Hydrolytic experiments of ThomasA mutants against IgA1$_{fl}$ (B) and wild-type ThomasA against IgA1$_{fl}$ mutants (C) using LC-MS. Color scale represents relative activity per mutation in ThomasA M64 peptidase domain and IgA1$_{CH1}$ from AF3 model of the complex, considering wild-type ThomasA as 100%. n.d. not detected. Results are expressed as mean ± SD from three independent biological replicates ($n = 3$). Source data are available online for this figure.

hydrolytic activity. Although this residue does not apparently interact with the CH1 domain of IgA1, it could interact with the antibody's light chain (Fig. EV4C). Mutations E486A, N516A, and Y519A in ThomasA also reduced the enzyme's activity. These residues would likely interact with key side chains in the CH1 domain—S161, H205, P212, S213—whose mutation also drastically reduced enzymatic activity. Interestingly, Y436A, which is close to the active site of ThomasA, did not affect the activity of the enzyme. Additionally, the S201A mutation in the CH1 domain, which might interact with Y514, significantly diminished enzyme activity. In the CH1 domain of the Fab region, most mutations impacted ThomasA activity. As a negative control due to its distant location from the interaction interface, the mutation D181A exhibited only a slight reduction in enzymatic activity compared to the wild-type Fab region. In the ThomasA-IgA1 AF3 model complex, C220 does not form disulfide bonds with its partner C196, as in the X-ray structure of the IgA1 Fab region (Fig. EV4). However, when C220

was mutated to serine, ThomasA was unable to hydrolyze the Fab, suggesting that this disulfide bridge needs to be formed for enzymatic activity. Interestingly, the N437A and W450A mutations in ThomasA, as well as the Q195A mutation in the CH1 domain, enhanced the enzyme's hydrolytic activity.

# Discussion

In this study, we provide structural insights into substrate recognition by M64 peptidases, focusing on ThomasA from *T. ramosa*, and BF3526 from *B. fragilis*, both bacteria found in the gut microbiome. Our SSN results show that the IgA specific protease ThomasA belongs to a small cluster (CL-V) among the M64 peptidase family. ThomasA is a ~130 KDa multidomain, monomeric protein that adopts an extended conformation, with a catalytic core solely responsible for IgA recognition. The X-ray

crystal structure of ThomasA[323-878] reveals that β1 and β2 domains are positioned opposite to the active site, suggesting a structural role rather than direct involvement in substrate recognition or catalysis. However, loop 7 of the β1 domain is located at one of the most distal ends of the catalytic site. The NTD and C-terminal β-sheet domains do not significantly impact catalytic efficiency, indicating that these domains may contribute to structural stability, regulation, mechanical pivoting, or cell surface localization (Liberato et al, 2016; Bogomolovas et al, 2016; Griffin et al, 2023; Lin et al, 2014; Tsuchiya et al, 2023).

Our functional data demonstrate that the catalytic core of ThomasA specifically recognizes the Fab region and the HR of both IgA1 and IgA2 but cannot hydrolyze the IgA1 HR independently or when presented in alternative protein scaffolds. Unlike previously described bacterial M26 and S6 IgA1 or IgG1 peptidases, which bind the Fc region either exclusively or in conjunction with the Fab, ThomasA follows a distinct substrate recognition mechanism, engaging the Fab region independently of Fc interactions (Wang et al, 2020; Vincents et al, 2004; Redzic et al, 2022; Johnson et al, 2009; Sudol et al, 2022). Other IgA-specific metallopeptidases, such as those from *S. pneumoniae* and *G. haemolysans* (M26 family), are also multidomain proteins. These enzymes engage both Fab and Fc regions of IgA1 through interactions with the NTD and CTD, respectively, and employ an active-site-gated mechanism that relies on substrate-induced conformational changes of the enzyme, maintaining a 1:1 stoichiometry (Wang et al, 2020; Redzic et al, 2022). In contrast, ThomasA appears to function via a distinct mechanism, likely relying solely on its M64 peptidase domain for IgA binding and cleavage, supporting a different binding stoichiometry. While not essential, the length and flexibility of the HR influence enzyme activity, as ThomasA exhibited greater hydrolytic activity toward longer Fab–HR variants than toward shorter IgA1 HR constructs or the naturally shorter IgA2 HR. Structural comparison of the active sites of ThomasA and BF3526 reveals that peptides containing IgA1 HR segments fit within the ThomasA active site but form only minimal interactions with the protein. This, along with the enzyme's inability to hydrolyze small peptides encoding the HR, supports the notion that ThomasA requires the presence of the Fab region for effective antibody processing.

ThomasA[323-878]-IgA1[CH1] AF3 model, combined with alanine scanning mutagenesis guided the identification of key residues in the enzyme and substrate essential for catalytic activity. In the CH1 of the Fab, we identified that residues in the β-strands F and G and their connecting loops were key for the interaction with the enzyme. The residues that had the most significant effect on ThomasA's hydrolytic activity are highly conserved in IgA1 and IgA2 but not in other antibody subclasses (Fig. EV5A). Mutations in ThomasA that affected its activity were located in loop 2, which contains the β-hairpin, α2, and β8. Interestingly, these regions are distant from the active site, suggesting that structural elements far from the catalytic center may interact with the Fab to position the HR correctly for cleavage. The amino acid composition of these structural elements was not conserved in BF3526 (Fig. EV5B) but was partially conserved in cluster CL-V of our SSN, where ThomasA is classified (Fig. EV5C). Additionally, we identified residues that positively modulate enzyme activity. These findings, together with the ability of ThomasA to efficiently hydrolyze IgA without being affected by glycosylation, provide a detailed molecular map of how ThomasA engages with IgA and suggest potential avenues for engineering enhanced enzyme variants for therapeutic applications.

We also expanded the determination of the substrate specificity of BF3526 from *B. fragilis* from M64 family. While ThomasA efficiently hydrolyzes IgA1 and IgA2, BF3526 exhibited distinct substrate specificity, suggesting different physiological roles within their respective bacterial species. These functional differences highlight the evolutionary plasticity of M64 peptidases, where structural modifications have led to specialization in substrate recognition and processing. Our enzymatic assays showed that BF3526 does not cleave intact IgA1 or IgA2 but hydrolyzes IgA fragments after pre-treatment with other IgAPs, within the N-terminal region. LC-MS analysis revealed a preference for cleavage sites following proline residues, consistent with its predicted catalytic mechanism and its specificity for the N-terminal region of IgA. The X-ray crystal structures of BF3526 in complex with substrate and product peptides support a single-displacement mechanism for catalysis of this enzyme and the M64 peptidase family. This mechanism is consistent with other metzincin metallopeptidases, including bacterial collagenases and matrix metalloproteinases (Cerdà-Costa and Gomis-Rüth, 2014; Dudev and Lim, 2000; Auld, 2013; Vallee and Auld, 1990; Bertini et al, 2006; Matthews and Matthews, 2002; Pelmenschikov and Siegbahn, 2002).

Structural comparison between ThomasA and BF3526 revealed a conserved M64 peptidase fold, but key differences in loop structures and active site accessibility may account for their divergent substrate specificities. In particular, BF3526 possesses an extended NTD β-sandwich, which may restrict substrate access to the active site. This structural constraint likely explains its inability to cleave intact IgA while permitting the accommodation of only two or three amino acid residues in the nonprime subsites. This domain could serve a regulatory function, controlling substrate engagement based on prior proteolytic processing. It is worth noting that our SSN analysis showed that almost 50% of M64 peptidases classified in Pfam database and contained in MEROPS belong to cluster CL-I, which includes BF3526. Proteins in this cluster, including BACOVA_00663 from *B. ovatus*, share high sequence similarity (39–99%), indicating a conserved substrate specificity distinct from the rest of the M64 family. Additionally, most proteins in this cluster are associated with bacteria from the order Bacteroidales, including the genera *Prevotella* and *Bacteroides*, which are commonly found in the human gut microbiota. These bacteria contain polysaccharide utilization loci (PULs) that encode the enzymatic machinery required to process complex carbohydrates (Martens et al, 2009). BF3526 and its homologues are not encoded in PULs and are not associated with other proteins involved in carbohydrate metabolism, suggesting that their primary function is independent of polysaccharide degradation. BF3526 contains a signal peptide indicative of secretion and given the abundance of hydrolytic enzymes in the gut microbiome, it may function in concert with other enzymes to degrade the N-terminal sequences of partially processed proteins. This cooperative activity could play a role in peptide turnover or nutrient acquisition within the gut environment (Kulkarni et al, 2024; Keller et al, 2023; Patra and Yu, 2022).

ThomasA is secreted by *T. ramosa*, formerly classified as *Clostridium ramosum*, a common enteric anaerobe that is generally

considered non-pathogenic, though it has been implicated in infections, particularly in young children and immunocompromised adults (Forrester and Spain, 2014). Additionally, this bacterium has been suggested to contribute to diet-induced obesity by enhancing nutrient absorption in gnotobiotic mice (Woting et al, 2014). Given the high abundance of IgA in the gastrointestinal and respiratory tract mucosa, several IgA1 proteases from pathogenic bacteria associated with meningitis have been shown to generate Fab fragments from specific IgA antibodies produced in vivo, facilitating bacterial adherence to mucosal surfaces and promoting cellular invasion (Kilian et al, 2022; Weiser et al, 2003). This raises the possibility that commensal bacteria, including *T. ramosa*, may utilize ThomasA as a mechanism to establish and maintain their niche within the gut environment. Moreover, the remarkable specificity of ThomasA for IgA antibodies has led to its exploration in animal models as a potential treatment for IgAN, an autoimmune disease for which no specific therapy is currently available (Xie et al, 2022). Notably, other bacterial peptidases have already been successfully introduced into clinical practice. For instance, IdeS (imlifidase), a cysteine protease, has been approved by the European Medicines Agency (EMA) for desensitizing highly sensitized patients awaiting kidney transplantation (Soveri et al, 2019; Jordan et al, 2017).

In summary, our study expands the understanding of M64 metallopeptidases by defining their structural diversity, substrate specificity, and catalytic mechanisms. Our findings reveal key differences in substrate recognition, domain contributions, and the impact of glycosylation, providing new insights into how these enzymes interact with IgA substrates and highlighting their functional diversity within the M64 family. The comparative analysis of ThomasA and BF3526 illustrates how structural variations influence enzymatic function, shedding light on their roles in bacterial physiology and their potential as biotechnological tools. Unraveling these mechanisms could enhance our understanding of how commensal bacteria interact with host immunity and contribute to microbial homeostasis. Additionally, this knowledge may inform the rational design of engineered enzymes with potential therapeutic applications in IgA-mediated autoimmune diseases.

# Methods

### Reagents and tools table

| Reagent/resource | Reference or source | Identifier or catalog number |
|---|---|---|
| **Experimental models** | | |
| Expi293 | Fisher Scientific | 13481059 |
| BL21(DE3) Competent *E. coli* | New England Biolabs | C2527H |
| XL10-Gold Ultracompetent Cells | Agilent Technologies | 200315 |
| **Recombinant DNA** | | |
| pGEX-4T-1 ThomasA[31-1167] from *T. ramosa* (UniProt code: Q9AES2, residues 31-1167) | ATG:biosynthetics GmbH | Custom |
| pSpeedET BF3526 from *B. fragilis* NCTC 9343 (UniProt code Q5L9L3, residues 19–426) | DNASU Plasmid Repository | BfCD00332456 |

| Reagent/resource | Reference or source | Identifier or catalog number |
|---|---|---|
| pcDNA3.1(+) IgA1$_{fl}$ | ATG:biosynthetics GmbH | Custom |
| pcDNA3.1(+) IgA1$_{HC}$ | ATG:biosynthetics GmbH | Custom |
| pcDNA3.1(+) IgA1$_{LC}$ | ATG:biosynthetics GmbH | Custom |
| MBP-TEV-IgA1$_{HR}$-EGFP | In house | Custom |
| **Antibodies** | | |
| Human IgA$_1$, Myeloma | Sigma-Aldrich® | 400109 |
| Anti-hEGFR-hIgA2 | ©InvivoGen | hegfr-mab7 |
| IgG1 | F. Hoffmann-La Roche Ltd | Herceptin (trastuzumab) |
| **Oligonucleotides and other sequence-based reagents** | | |
| PCR primers | This study | Appendix Table S1 |
| VPCPVPSTPP | Biomatik | Custom |
| **Chemicals, enzymes and other reagents** | | |
| IgASAP Sub1 | Genovis | I0-IA1-010 |
| IgASAP Sub1 + 2 | Genovis | I0-IA3-010 |
| OmniGLYZOR | Genovis | G3-OM6-005 |
| PNGase F | Genovis | G1-PF6-010 |
| GalactEXO | Genovis | G1-GM1-020 |
| SialEXO | Genovis | G1-SM1-020 |
| TEV (Tobacco Etch Virus Protease) | In house | |
| CX35 | Sastre DE et al, 2024 | |
| Invitrogen™ Luria Broth Base (Miller's LB Broth Base), powder | Fisher Scientific | 11558866 |
| Tris Base | Fisher BioReagents™ | 10376743 |
| Sodium chloride | Sigma-Aldrich® | S9625 |
| Benzonase® Nuclease | Merck | 70746-4 or 71205-3 |
| Pierce™ Protease Inhibitor Mini Tablets, EDTA-free | Thermo Scientific™ | A32955 |
| L-Glutathione reduced | Merck | G6529 |
| Imidazole | Fisher Scientific | 10561331 |
| Phosphate-Buffered Saline, 1X without calcium and magnesium, PH 7.4 ± 0.1 | Corning® | 21-040-CV |
| Glycine | Fisher Scientific | 12317153 |
| Zinc chloride | Sigma-Aldrich® | Z0152 |
| **Software** | | |
| Enzyme Function Initiative: EFI-EST | https://efi.igb.illinois.edu/ Oberg N et al, 2023 | |
| Cytoscape | https://cytoscape.org/ Shannon et al, 2003 | |
| WebLogo 3 | https://weblogo.threeplus one.com/create.cgi https://weblogo.berkeley. edu/logo.cgi Crooks GE et al, 2004 | |
| ScÅtter | Rambo and Tainer, 2011 | |

| Reagent/resource | Reference or source | Identifier or catalog number |
|---|---|---|
| PRIMUS | Konarev PV et al, 2003 | |
| GNOM | Svergun DI, 1992 | |
| GASBOR | Svergun DI et al, 2001 | |
| SAXSMoW | Piiadov V et al, 2019 | |
| CRYSOL | Svergun D et al, 1995 | |
| CryoSPARC v4.6.0 | Punjani A et al, 2017 | |
| Phaser | McCoy AJ et al, 2007 | |
| PHENIX suite | Adams PD et al, 2010 | |
| COOT | Emsley P et al, 2010 | |
| AlphaFold | Jumper et al, 2021 | |
| AF3 | Abramson J et al, 2024 | |
| UCSF Chimera | Kabsch W, 2010 | |
| DALI | Holm L et al, 2023 | |
| PISA | Krissinel and Henrick, 2007 | |
| Clustal Omega server | Madeira F et al, 2024 | |
| ESPRIPT 3.0 web tool | Robert and Gouet, 2014 | |
| BioConfirm10.0 | https://www.agilent.com/en/product/software-informatics/mass-spectrometry-software/data-analysis/bioconfirm-software | |
| MSFragger | https://msfragger.nesvilab.org/ | |
| **Other** | | |
| Corning™ SFCA syringe filters | Fisher Scientific | 10002471 |
| High-resolution GST-tagged protein purification with GSTrap™ HP | Cytiva | 17528201 |
| HisTrap™ HP His tag protein purification columns | Cytiva | 17524802 |
| HiTrap™ Protein A HP antibody purification column | Cytiva | 17040301 |
| Superdex 200 Increase 10/300 GL | Cytiva | 28990944 |
| ÄKTA Pure™ | Cytiva | |
| Pierce™ Glutathione Agarose | Thermo Scientific™ | 16100 |
| Ni-NTA Agarose | Qiagen | 30210 |
| CaptureSelect™ IgA-XL Affinity Matrix | Thermo Scientific™ | 2943972005 |
| SurePAGE™, Bis-Tris, 10×8, 4–20%, 10 wells | GenScript | M00655 |
| InstantBlue® Coomassie Protein Stain (ISB1L) | Abcam | ab119211 |
| Amicon® Ultra Centrifugal Filter, 30 kDa MWCO | Merck | UFC903008 |
| Carbon Film 300 Mesh, Ni | Electron Microscopy Sciences | CF300-Ni-50 |
| BioResolve RP mAb polyphenyl column | Waters | 186008945 |
| Acquity Premier CSH C18 1.7 µm, 2.1 ×150 mm column | Waters | 186009462 |

## Sequence similarity network analysis of M64 family from Pfam database

We constructed sequence similarity networks (SSN) using all annotated proteins in M64 metallopeptidase family from the Pfam database (PF16217 and PF09471) using the Enzyme Similarity Tool (EFI-EST) (Gerlt et al, 2015) (https://efi.igb.illinois.edu/), selecting an alignment score threshold of 50 (in order to reach an alignment stringency that allowed separation into distinct clusters in which proteins shared at least sequence identity of 30%). Then, we used Cytoscape software (Shannon et al, 2003) (ttps://cytoscape.org/)) for deep analysis of the resulting SSNs. Nodes were organized with the Cytoscape Prefuse Force Directed Open CL layout %id, and colored by neighborhood connectivity using EFI-EST (Oberg et al, 2023). Neighborhood connectivity coloring measures local network interconnectivity for detection of protein families within SSN clusters. The network comprises the amino acid sequence of 6830, expressed as 4573 representative nodes with a total of 8,130,269 edges between pairs of representative nodes, indicating a Blast $-\log(E\text{-value})$ of 5 or better. The convergence ratio value for the entire family is 0.778. The convergence ratio is a measure of the similarity of the sequences used in the BLAST. It is the ratio of the total number of edges retained from the BLAST (e-values less than the specified threshold ( = 5). The convergence ratio can be used as a criterion to infer whether an SSN cluster is isofunctional. The value decreases from 1.0 for sequences that are identical to 0.0 for completely unrelated sequences. SSN analysis was made using the full-length sequence of M64 family proteins, however, this analysis is based on a local alignment that generates clusters based on similarity of protease domains. Consensus sequences for selected protein regions of each cluster were determined by using WebLogo 3 (Crooks et al, 2004) (https://weblogo.threeplusone.com/create.cgi or https://weblogo.berkeley.edu/logo.cgi).

## Cloning, expression and purification of ThomasA constructs and mutants

ThomasA[31-1167] from *T. ramosa* (UniProt code Q9AES2, residues 31-1167) was cloned into a pGEX-4T-1 vector using BamHI and XhoI sites (ATG:biosynthetics GmbH, Mezhausen, Germany). This vector was used as a template to generate the ThomasA constructs ThomasA[31-878], ThomasA[31-795], ThomasA[31-632], ThomasA[323-632] and ThomasA[323-878] by the FastCloning method (Li et al, 2011) (Appendix Table S1). The single point mutants of ThomasA[323-878] were generated by QuickChange site-directed mutagenesis (Liu and Naismith, 2008) (Appendix Table S1). All ThomasA constructs, fused to GST tag protein, were produced in *Escherichia coli* BL21(DE3) competent cells (New England Biolabs). Culture cells were grown in 2 L of LB medium supplemented with 100 µg mL$^{-1}$ of ampicillin (AMP) at 37 °C. When the culture reached an $OD_{600}$ of 0.6–0.8, expression was induced by adding

1 mM isopropyl β-D-1-thio-galactopyranoside (IPTG). After ca. 16–20 h at 18 °C, cells were harvested by centrifugation at 5000 × g for 20 min at 4 °C and resuspended in 40 mL of buffer A (50 mM Tris-HCl pH 7.5, 500 mM NaCl), 0.8 μL of benzonase nuclease (Merck) and an EDTA free tablet of protease inhibitors (A32955, ThermoFisher). Cells were disrupted by sonication (18 cycles of 10 s pulses with 60 s cooling intervals between the pulses, and 60% of amplitude) in ice, and the suspension was centrifuged at 50,000 × g for 30 min at 4 °C. For ThomasA$^{31-1167}$, ThomasA$^{31-878,}$ and ThomasA$^{323-878}$, the supernatant was filtered with 0.45 μm filters and then applied into a GSTrap HP column (5 mL, Cytiva) equilibrated in buffer A. The elution was performed in an ÄKTA pure™ (Cytiva) using a linear gradient from 0 to 100% of buffer A with 10 mM L-glutathione reduced (Merck) for 20 min at 5 mL min$^{-1}$. The protein was dialyzed against buffer B (50 mM Tris-HCl, pH 7.5, 150 mM NaCl), with TEV protease (1:20 w/w ratio), overnight at 4 °C. The completeness of the enzymatic digestion reaction was confirmed by 4–20% SDS-PAGE gels (GenScript) and the bands were visualized by InstantBlue® Coomassie Protein Stain (abcam). The protein was loaded into a GSTrap HP column (5 mL, GE Healthcare) equilibrated with buffer B. Flow-through was loaded into a HisTrap HP (5 mL, GE Healthcare), previously equilibrated in buffer B. Next, the flow-through was concentrated (Amicon Ultra 30 K, Merck) and purified by size-exclusion chromatography using a HiLoad 10/300GL Superdex 200 column (Cytiva) equilibrated with buffer B for SAXS and hydrolytic experiments, and buffer C (20 mM Tris-HCl pH 7.5) for X-ray crystallography experiments. For SAXS and hydrolytic experiments by SDS-PAGE, ThomasA constructs were concentrated again, flash-frozen and stored at −80 °C until use. ThomasA constructs purity was analyzed by 4–20% SDS-PAGE gels (GenScript) and the bands were visualized by InstantBlue® Coomassie Protein Stain (abcam), and an average of 3 mg per L of culture was obtained.

For hydrolytic experiments by LC-MS, control wild-type ThomasA$^{323-878}$ and single-point mutants were purified with slight modifications. After obtaining the filtered lysate, it was incubated with GST beads (16100, Thermo Scientific) pre-equilibrated in buffer A at 18 °C for 3 h. The protein was then eluted using 10 mM L-glutathione reduced (Merck) in buffer A and dialyzed overnight at 4 °C against the buffer B, with TEV protease (1:20 w/w ratio). The completeness of the enzymatic digestion reaction was confirmed by SDS-PAGE. Next, the protein was incubated 5 min with Ni-NTA agarose resin (Qiagen) pre-equilibrated in buffer B. The flow through was concentrated (Amicon Ultra 30 K, Merck) and loaded in GST beads (16100, Thermo Scientific), pre-equilibrated in buffer B, at 18 °C for 3 h. Finally, flow through was concentrated (Amicon Ultra 30 K, Merck), flash-frozen and stored at -80 °C until use. On average, 0.5 mg of pure protein per L of bacterial culture was obtained.

## Cloning, expression and purification of BF3526 constructs

BF3526 from *B. fragilis* NCTC 9343 (UniProt code Q5L9L3, residues 19–426) in pSpeedET vector was purchased from DNASU Plasmid Repository (https://dnasu.org/DNASU/Home.do). This vector was used as a template to generate the BF3526$^{161-426}$ using the FastCloning method (Li et al, 2011) (Appendix Table S1). BF3526 constructs, fused to a His-Tag, were expressed in *E. coli*

BL21(DE3) competent cells (New England Biolabs). Cultures were grown in LB medium supplemented with 100 μg mL$^{-1}$ of AMP at 37 °C to an OD$_{600}$ of 0.6–0.8, at which point the cultures are cooled down to 18 °C to induce the expression with 1 mM IPTG and incubated at 18 °C for 16–20 h. Cells were harvested by centrifugation at 5000 × g for 20 min at 4 °C and resuspended in 50 mL of buffer A, 3 μL of benzonase nuclease (Merck) and an EDTA-free tablet of protease inhibitors (A32955, ThermoFisher). Cells were lysed by sonication in ice (18 cycles of 10 s pulses with 60 s cooling intervals between the pulses, and 60% amplitude). The lysate was centrifuged at 50,000 × g for 30 min at 4 °C. The supernatant was filtered through 0.45 μm filters and then applied into a HisTrap HP column (5 mL, Cytiva) equilibrated with buffer A. The elution was performed in an ÄKTA pure™ (Cytiva) using a linear gradient from 0 to 100% of buffer A with 500 mM imidazole (Fisher) for 20 min at 5 mL min$^{-1}$. The protein was dialyzed against the dialysis buffer B with TEV protease (1:20 w/w ratio), overnight at 4 °C. The protein was loaded into a HisTrap HP (5 mL, GE Healthcare) equilibrated in buffer B. Next, the flow-through was concentrated (Amicon Ultra 30 K, Merck) and purified by size-exclusion chromatography using a HiLoad 16/600 Superdex 200 column (Cytiva) equilibrated with buffer B for hydrolytic experiments, and buffer C for X-ray crystallography experiments. For hydrolytic experiments, the different constructs were concentrated again, flash-frozen and stored at −80 °C until use. BF3526 constructs purity was analyzed by 4–20% SDS-PAGE gels (GenScript) through the purification process and the bands were visualized by InstantBlue® Coomassie Protein Stain (abcam). An average of 12 mg per L of culture was obtained.

## Cloning, expression and purification of IgA constructs and mutants

The heavy chain variable domain (VH) (PDB code 7K75) (Tan et al, 2021), preceded by a Kozak sequence ACCATGG and along with CH1, HR, CH2 and CH3 domains of IgA1$_{fl}$ (Uniprot code P01876, residues 1-335) were cloned in a pcDNA3.1 vector using BmHI and XbaI sites (ATG:biosynthetics GmbH, Mezhausen, Germany), hereinafter referred to as IgA1$_{HC}$. This vector was used as a template to generate the IgA1 constructs IgA1$_{Fab-HR}$, IgA1$_{Fab-PVPS}$, IgA1$_{Fab-HR}$-IgG1$_{Fc}$ and IgG1$_{Fab}$-IgA1$_{HR-Fc}$ by the FastCloning method (Li et al, 2011) (Appendix Table S1). Single-point mutants of IgA1$_{HC}$ were generated by QuickChange site-directed mutagenesis (Liu and Naismith, 2008) (Appendix Table S1). Similarly, the light chain variable (VL) and constant (CL) domains (PDB code 7K75) (Tan et al, 2021) were cloned in the same vector sites (ATG:biosynthetics GmbH, Mezhausen, Germany), hereinafter referred to as IgA1$_{LC}$. The IgA1$_{HC}$ and IgA1$_{LC}$ plasmids were transfected in Expi293 cells using the manufacturer's protocol (MAN0007814, ThermoFisher Scientific) in a ratio 1:1 with the addition of Penicillin/Streptomycin mix 24 h after transfection to produce IgA1$_{fl}$. The cells were cultured for 96 h before harvesting. The IgA1$_{fl}$ constructs and mutants were purified using Capture-Select™ IgA-XL Affinity Matrix (2943972005, ThermoFisher Scientific) with PBS pH 7.4 being used as the binding buffer and 100 mM sodium citrate buffer pH 3.0 as the elution buffer. The fractions were neutralized with 1 M Tris-HCl pH 9.3. SDS-PAGE was used to assess protein purity. The proteins were concentrated, flash-frozen and stored at −80 °C until ready for use.

The IgA1-IgG1 chimeras were produced using the FastCloning method (Li et al, 2011) (Appendix Table S1): (i) by cloning the Fd and HR of IgA1 between residues 1-240 into the pcDNA3.1 vector containing CD4-induced the full length heavy chain of IgG1 (Guan et al, 2013), maintaining the scFc of IgG1 (IgA1$_{Fab-HR}$-IgG1$_{Fc}$), and (ii) by cloning the Fd of IgG1 between residues 1-103 into the pcDNA3.1 vector containing the IgA1$_{fl}$, maintaining the HR and the scFc of IgA1 (IgG1$_{Fab}$-IgA1$_{HR-Fc}$). The vector containing IgA1$_{Fab-HR}$-IgG1$_{Fc}$ and the IgG1$_{Fab}$-IgA1$_{Fc-HR}$ were transfected with the IgA1$_{LC}$ and the IgG1$_{LC}$ vectors (Guan et al, 2013), respectively, identically as IgA1$_{fl}$. IgG1$_{Fab}$-IgA1$_{Fc-HR}$ was purified as IgA1$_{fl}$ and IgA1$_{Fab-HR}$-IgG1$_{Fc}$ was purified using Protein A chromatography with PBS pH 7.4 as the binding buffer and 100 mM sodium citrate buffer pH 3.0 as the elution buffer. The fractions were neutralized with 1 M Tris-HCl pH 9.3. SDS-PAGE was used to assess protein purity. The proteins were concentrated, flash-frozen and stored at -80 °C until ready for use.

The IgA1$_{Fab-HR}$, IgA2$_{Fab-HR}$, and IgA1$_{Fab-PVPS}$ constructs, each containing a C-terminal His-tag, were expressed in Expi293 cells as previously described. Following expression, the culture supernatant was harvested and clarified by filtration through 0.22 μm filters. The filtered supernatant was incubated overnight at 4 °C with Ni-NTA agarose resin (Qiagen) pre-equilibrated with PBS (pH 7.4). Bound proteins were eluted using 250 mM imidazole (Thermo Scientific) in PBS. The eluted proteins were subsequently buffer-exchanged into PBS (pH 7.4), flash-frozen in liquid nitrogen, and stored at −80 °C until further use.

Gd-IgA1$_{fl}$ and Gd-IgA1$_{Fab-HR}$ were generated by treating IgA1$_{fl}$ and IgA1$_{Fab-HR}$ with GalactEXO and SialEXO (Genovis), following the manufacturer's instructions.

The HR of IgA1 (Uniprot code P01876, residues 100–121) was cloned between the MBP-TEV and EGFP tags of MBP-TEV-EGFP vector (Sheffield et al, 1999) using the FastCloning method (Li et al, 2011) (Appendix Table S1), hereinafter referred to as MBP-TEV-IgA1$_{HR}$-EGFP. The expression and purification protocols follow the same procedures as those described for BF3526 above.

## SEC-SAXS experiments of ThomasA

Small-Angle X-ray Scattering coupled with Size Exclusion Chromatography (SEC-SAXS) data for recombinant purified ThomasA$^{31-1167}$, ThomasA$^{31-878}$, or ThomasA$^{323-878}$ in 50 mM Tris-HCl pH 7.5, 100 mM NaCl, 2% v/v glycerol were collected on the B21 beamline of the Diamond Light Source, UK. Data were collected using an EigerX 4 detector (Dectris, CH) at a sample-detector distance of 3692.9 mm and a wavelength of $\lambda = 0.9464$ Å. The range of momentum transfer of $0.1 < s < 5$ nm$^{-1}$ was covered ($s = 4\pi\sin\theta/\lambda$, where $\theta$ is the scattering angle). In total, 50 μL of a protein sample at 2 mg mL$^{-1}$ were injected into a Superdex 200 Increase 3.2 column and eluted at a flow rate of 75 μL min$^{-1}$. Data were processed and merged using standard procedures by the program package ScÅtter (Rambo and Tainer, 2011) and PRIMUS (Konarev et al, 2003). The maximum dimensions ($D_{max}$), the interatomic distance distribution functions ($P(r)$), and the radius of gyration ($R_g$) were computed using GNOM (Svergun, 1992). The ab initio SAXS envelopes of ThomasA$^{31-1167}$, ThomasA$^{31-878}$, and ThomasA$^{323-878}$ were calculated using GASBOR (Svergun et al, 2001). The results and statistics are summarized in Appendix

Table S2. The molecular weight from the SAXS data was calculated using SAXSMoW (Piiadov et al, 2019). Using CRYSOL (Svergun et al, 1995), we fitted ThomasA$^{323-878}$ SAXS data into the solution-scattering profile calculated from the crystal structure (PDB code 9QDH). The SAXS data of ThomasA have been deposited into the Small Angle Scattering Biological Data Bank, under the accession codes SASDX45 (ThomasA$^{31-1167}$), SASDX55 (ThomasA$^{31-878}$) and SASDX65 (ThomasA$^{323-878}$).

## Negative stain EM

Purified ThomasA$^{31-878}$ was used for standard negative stain grid preparation using continuous glow discharged carbon EMgrids CF300-Ni (EMS, USA). Briefly, drops of 8 μL of purified sample at 10 μg mL$^{-1}$ were placed on the grid, incubated for 90 s and then blotted to remove the excess with filter paper (Whatman®). The grids were washed by placing them onto a drop of PBS, pH 7.4 for 30 s and bottled dry. The complex was stained by transferring the grid to a drop of 1% w/v uranyl formate for 45 s, then dried with a filter paper and stored at room temperature. Samples were imaged at ×120,000 magnification using a Talos L120C 120 kV D5550 CryoTwin equipped with an BM-Ceta camera (Gatan, USA) from the Robert P. Apkarian Integrated Electron Microscopy Core at Emory University, GA. Ca 150 Images were used to analyze particles by single particle analysis using CryoSPARC v4.6.0 (Punjani et al, 2017).

## Hydrolytic activity assays of ThomasA against IgA constructs by LC-MS

Time-course and endpoint hydrolytic activity assays were performed to evaluate the substrate specificity of ThomasA under comparable buffer conditions. For time-course hydrolytic activity assays, 5 μM of IgA1$_{fl}$ or IgG1$_{Fab}$-IgA1$_{Fc-HR}$ were mixed with 0.28 μM of ThomasA$^{31-1167}$, ThomasA$^{31-878}$, or ThomasA$^{323-878}$ in 50 mM Tris-HCl pH 7.5, 150 mM NaCl, and 0.5 mM ZnCl$_2$, in a total volume of 30 μL. Reactions were incubated in the LC-MS at 37 °C, with aliquots sampled approximately every ~6.5 min until completion. In contrast, endpoint assays for IgA1 constructs, IgA2$_{Fab-HR}$, IgG1$_{Fab}$-IgA1$_{HR-Fc}$ and IgA1$_{fl}$ single-point mutants were carried out using 0.33 μM of ThomasA constructs and 6 μM of substrate under the same buffer and in a total volume of 20 μL. These reactions were stopped by adding a 10 μL of a mixture of 15 mM EDTA and 0.07% formic acid after 45 min at 37 °C. All reactions were performed in triplicate and analyzed by LC-MS using an Agilent 1290 Infinity II LC System equipped with a 50 mm PLRP-S column (1000 Å pore size, Agilent). The LC system is coupled to an Agilent 6545XT quadrupole-time of flight (Q-TOF) mass spectrometer (Agilent, Santa Clara, CA). Relative amounts of the substrate and hydrolysis product were quantified after deconvolution of the raw data, and identification of the corresponding peaks using BioConfirm10.0 (Agilent, Santa Clara, CA). For IgA2$_{Fab-HR}$, the $N$-glycans on the CH1 domain were previously removed using CX35 (Sastre et al, 2024). For IgA1, Gd-IgA1, IgG1$_{Fab}$-IgA1$_{HR-Fc}$, and IgA1$_{fl}$ single-point mutants, quantification was performed before deconvolution of the raw data due to the peak separation between the Fab region and the full-length antibody and the poor ionization of the intact substrates.

## ThomasA[323-878] and BF3526 crystallization and data collection

The crystal of ThomasA[323-878] was obtained by mixing 0.25 μL of the protein (12.2 mg mL$^{-1}$) in buffer C and 5 mM of VPCPVPSTPP (Biomatik) with 0.25 μL of mother liquor containing 25% (w/v) polyethylene glycol (PEG) 3350, 100 mM Tris-HCl pH 8.5 and 200 mM NaCl (Index Screen HT protein crystallization screen, Hampton Research). The crystal appeared after 21 days and was washed with the mother liquid and 30% (v/v) of glycerol and frozen under liquid nitrogen. BF3526 crystal was obtained by mixing 0.25 μL of the protein (10 mg mL$^{-1}$) in buffer C and 5 mM of VPCPVPSTPP (Biomatik), with 0.25 μL of mother liquor containing 0.1 M Tris Bicine pH 8.5, 20% ethylene glycol, 10% PEG 8000, 0.12 M 1,6-hexanediol, 0.12 M 1-butanol, 0.12 M 1,2-propanediol, 0.12 M 2-propanol, 0.12 M 1,4-butanediol and 0.12 M 1,3-propanediol (Morpheus, Molecular Dimensions). The crystal appeared after 12 h and was directly frozen under liquid nitrogen. ThomasA and BF3526 X-ray diffraction data were collected on a EIGER X 9 M photon-counting area and PILATUS3 6 M detectors, respectively, both at the microfocus I24 beamline (Diamond Light Source (DLS), UK, see Appendix Table S3 for details). Data were integrated and scaled with XDS, following standard procedures (Kabsch, 2010).

## ThomasA[323-878] and BF3526 structures determination and refinement

Structural determination of ThomasA[323-878] and BF3526 was solved using molecular replacement with Phaser (McCoy et al, 2007) and PHENIX suite (Adams et al, 2010), by employing AF template (AF-Q9AES2-F1) (Jumper et al, 2021; Varadi et al, 2024), and the previously reported BF3526 unliganded structure (PDB code 4DF9), respectively. Several rounds of refinement using Phenix (Liebschner et al, 2019) were alternated with model building using COOT (Emsley et al, 2010). The final processing and refinement statistics of the apo ThomasA[323-878] and BF3526 datasets are reported in Appendix Table S3. Atomic coordinates and structure factors have been deposited into the Protein Data Bank, with accession codes 9QDH (ThomasA[323-878]) and 9QDI (BF3526).

## BF3526 fragmentation and LC-MS analysis

BF3526[NTD], BF3526[CT1] and BF3526[CT2] fragments (BF3526[D]) were obtained by incubating purified BF3526 with chymotrypsin at a 1:0.02 ratio in 50 mM Tris-HCl pH 7.5, 150 mM NaCl, and 0.5 mM ZnCl$_2$ at 37 °C, 300 rpm, for 18–20 h. BF3526[D] was loaded onto a 4–20% SDS-PAGE gels (GenScript) and the bands were visualized by InstantBlue® Coomassie Protein Stain (abcam). Excised SDS-PAGE bands were washed and proteins digested using a standard in-gel digestion procedure. Briefly, proteins were reduced (10 mM TCEP, 45 min, 56 °C), alkylated (55 mM CAA, 30 min, RT in the dark) and digested with trypsin (Promega) (estimated enzyme:protein ratio 1:100) in 5% ACN, 1 mM CaCl$_2$, 100 mM ammonium bicarbonate for 16 h at 37 °C. Digestion was stopped with formic acid to a final concentration of 5%. Resulting peptides were desalted using C18 stage-tips. Peptides were analyzed by LC-MS/MS analysis using a Vanquish Neo LC system (Thermo) coupled to an Exploris 480 mass spectrometer (Thermo). Raw files were

analyzed with MSFragger against an *E. coli* protein database (Uniprot) supplemented with the actual BF3526 protein sequence. Trypsin-mediated fragmentation of BF3526 was carried out at the Proteomics Platform of IIS Biobizkaia (Spain).

## Hydrolytic activity assays by SDS-PAGE

Recombinant IgA1[fl], human IgA1 from myeloma (hIgA1) (100 μg; Calbiotech), human anti-hEGFR IgA2 (hIgA2) (50 μg; Invivogen), or IgG1 (trastuzumab (Roche)) were incubated with BF3526 or ThomasA[323-878] at a 1:1 enzyme-to-substrate ratio (1.5 μM). MBP-TEV-IgA1HR-EGFP or hIgA1 were incubated with ThomasA[31-1167], ThomasA[31-878] or ThomasA[323-878] in identical conditions. BF3526 was also incubated with albumin and BF3526[NTD], BF3526[CT1] and BF3526[CT2] fragments (BF3526[D]) were incubated with IgA1[fl], and hIgA2 or albumin, at the same reaction conditions. All reactions were incubated in 50 mM Tris-HCl pH 7.5, 150 mM NaCl, and 0.5 mM ZnCl$_2$ at 37 °C and 300 rpm, for 18–20 h. Reactions were stopped by adding 5X SDS-PAGE loading buffer (50% glycerol, 10% SDS, 500 mM DTT, 250 mM Tris-HCl pH 6.8, 0.5% bromophenol), and proteolytic activity was assessed by detecting qualitative molecular weight changes in IgA substrates via 4–20% SDS-PAGE (GenScript) analysis. The bands were visualized by InstantBlue® Coomassie Protein Stain (Abcam).

## Hydrolytic activity of BF3526 towards pre-digested IgA1 and IgA2 analyzed by LC-MS

hIgA1 was digested using IgASAP Sub1 (PSTPP/TPSPS) and Sub1 + 2 (TVPCP/VPSTP; Genovis), and hIgA2 was digested using IgASAP Sub1 + 2 (TVPCP/VPPPP). Digestions were performed using 1 U of enzyme/μg of IgA at 2 mg mL$^{-1}$ of IgA in PBS pH 7.4. Reactions were performed at 37 °C, 1 h for IgASAP Sub1 and 18 h for IgASAP Sub1 + 2. To facilitate the analysis, the samples were deglycosylated using OmniGLYZOR (Genovis) which constitutes a mixture of immobilized enzymes for complete deglycosylation of *N*-glycans and simple core 1 *O*-glycans. Samples were applied to the equilibrated OmniGLYZOR columns and incubated at 37 °C for 4 h in a thermal shaker mixing at 750 rpm. To remove all *N*-glycans, lyophilized PNGase F (Genovis) was used under reducing and denaturing conditions. DTT and lauroylsarcosine were added to the samples to final concentrations of 50 mM and 0.5%, respectively, and heated to 90 °C for 5 min. After allowing the samples to cool down, PNGase F was applied at 1 U/μg of IgA and incubated at 50 °C for 30 min. Removal of reducing and denaturing agents was performed using two subsequent Zeba spin desalting columns 40k MWCO, 0.5 mL (Thermo Scientific), according to manufacturer's instructions. Half of the samples were used as controls and stored at -20 °C until analysis. The other half of the samples were subjected to BF3526 digestion using an enzyme-to-substrate ratio of 1:10 (wt:wt). Reactions were performed at 37 °C for 18 h. Samples were reduced and denatured at 37 °C for 30 min using 100 mM DTT and 4 M guanidine-HCl and analyzed by LC-MS on an Agilent LC 1290 Infinity system controlled by the Compass HyStar software (version 5.1.8.1, Bruker) and coupled to a Bruker Impact II ESI-QTOF MS controlled by the Control software (version 5.2, Bruker). Subunits were separated by reversed-phase LC on a BioResolve RP mAb polyphenyl, 450 Å, 2.7 μm, 2.1 ×100 mm column (Waters) at 0.2 mL min$^{-1}$ flow using the following gradient: 10% B for 20 min,

10–31% B in 2 min, 31–55% B in 10 min, 55–90% B in 1 min, 90% for 1 min. Mobile phase A: 0.1% formic acid in ultrapure $H_2O$; mobile phase B: 0.1% formic acid in 95% ACN. The column temperature was kept at 80 °C. For each analytical run, 1 µg of antibody was used. The MS instrument was operated in positive mode with a capillary voltage of 4.5 kV and the needle temperature 220 °C. Mass spectrometric data were acquired in the *m/z* range 300–3000. Data were processed and deconvoluted using the Compass Data Analysis software (version 5.2, Bruker).

## Hydrolytic activity of BF3526 towards hIgA2 and IgG1 peptides analyzed by LC-MS/MS

For the generation of hIgA2 trypsin-digested peptides, 35 µg of human hIgA2 at 1 mg mL$^{-1}$ was reduced and denatured using 5 mM DTT and 5.5 M guanidine at 37 °C for 60 min. Alkylation was performed by incubation with 10 mM iodoacetamide in the dark at room temperature for 30 min. The sample was buffer-exchanged to PBS using Zeba spin desalting columns 40k MWCO, 0.5 mL before trypsin was added at 1:20 (mol:mol) and allowed to react at 37 °C for 18 h. Trypsin was heat-inactivated and the sample centrifuged at 16,000×*g* for 10 min. The supernatant was saved and was either used as control or digested using BF3526 for 2 h or 18 h (1:10; ~1 mg mL$^{-1}$ hIgA2) at 37 °C. Reactions were quenched by the addition of formic acid to a final concentration of 0.1%. For the generation of IgG1 Lys-C-digested peptides, 125 µg of trastuzumab (Roche) at 10 mg mL$^{-1}$ was reduced and denatured using 5 mM DTT and 4 M guanidine at 37 °C for 60 min. Alkylation was performed by incubation with 15 mM iodoacetamide in the dark at 37 °C for 30 min. rLysC (Promega) was added to the sample (1:50, wt:wt) followed by incubation at 37 °C for 60 min. LysC-digested peptides were purified using C18 tips according to the manufacturer's instructions (Thermo Scientific) and purified peptides were speed vacuum centrifuged to dryness. The dried peptides were dissolved in PBS and divided into three samples:control, or further digestion using BF3526 for 2 h or 18 h (1:10 (wt:wt); ~1 mg mL$^{-1}$ IgG1) at 37 °C. Reactions were quenched by the addition of formic acid to a final concentration of 0.1%. All samples were analyzed by LC-MS on an Agilent LC 1290 Infinity system controlled by the Compass HyStar software and coupled to a Bruker Impact II ESI-QTOF MS. Peptides were separated by revered-phase LC on a Acquity Premier CSH C18 1.7 µm, 2.1 ×150 mm column (Waters) at 0.2 mL min$^{-1}$ flow using the following gradient: 2% B for 5 min, 2–45% B in 40 min, 45–90% B in 2 min, 90% for 2 min. Mobile phase A: 0.1% formic acid in ultrapure $H_2O$; mobile phase B: 0.1% formic acid in 95% ACN. The column temperature was kept at 50 °C. For each analytical run, 3 µg or 5 µg of sample was used. The MS instrument was operated in positive mode with a capillary voltage of 4.5 kV and the needle temperature 200 °C. Mass spectrometric data were acquired in the *m/z* range 150–4000. Data were processed using the Compass Data Analysis software and peptide searches were performed using the BioPharma Compass 2021 software (version 4.0.0 742, Bruker). Searches were performed against trastuzumab with no enzyme, allowing for up to 50 missed cleavages, fixed carbamidomethyl modifications of Cys, MS tolerance of 20 ppm and MS/MS tolerance of 0.025 Da. For data presentation, only peptides where BF3526 digestion was observed were included.

## Hydrolytic activity of BF3526 and ThomasA$^{323-878}$ towards VPCPVPSTPP analyzed by LC-MS/MS

The peptide VPCPVPSTPP was dissolved in PBS (10 mM) and mixed with BF3526 at different enzyme-to-substrate ratios (mol:mol): 1:100, 1:1000, and 1:10,000 and with ThomasA$^{323-878}$ at 1:100 ratio. Reactions were incubated at 37 °C for 0–18 h and quenched using 0.1% formic acid. Samples were analyzed using the corresponding LC-MS method as described in the previous section with the following exceptions: the gradient used was 2% B for 4 min, 2–60% B in 14 min, 60–90% B in 1 min, 90% for 1 min, the column temperature was kept at 40 °C, and 1 µg of sample was used for each analytical run.

## Analysis of ThomasA$^{323-878}$ and IgA1$_{Fab-HR}$ complex formation

The purified inactive ThomasA$^{323-878}$-H539A/E540A (5 µM) and IgA1$_{Fab-HR}$ (25 µM) were mixed in a 1:5 ratio in 20 mM Tris pH 7.5 and 150 mM NaCl and incubated for 30 min at RT. The mixture was then applied to a HiLoad 10/300GL Superdex 200 column (Cytiva). For crosslinking, inactive ThomasA$^{323-878}$ (5 µM) and IgA1$_{Fab-HR}$ (25 µM) were incubated in PBS for 30 min at RT. Crosslinking reagents were then added to the mixture: disuccinimidyl suberate (DSS), ethylene glycol bis(succinimidyl succinate) (EGS), and bis(sulfosuccinimidyl)suberate (BS3) to a final concentration of 1 mM, and glutaraldehyde (GA) to a final concentration of 0.25%. Complex formation was assessed by SDS-PAGE.

## Structural analysis and sequence alignments

Predicted structure models for individual proteins and ThomasA$^{323-878}$-IgA1$_{CH1}$ complex were obtained using AF3 (Abramson et al, 2024). Structure-based sequence alignment analysis, molecular graphics and volume rigid-body fitting were performed using UCSF Chimera (Pettersen et al, 2004). Moreover, structural alignment of the M64 peptidase domain of ThomasA and BF3526 was performed using the Chimera Match and Align tool (residue-residue distance cutoff of 10 Å), and electrostatic surfaces were represented by using the Coulombic Surface Coloring tool. Structural homologues search for protein domains and Z-score values were obtained from DALI (Holm et al, 2023). The buried area of ThomasA$^{323-878}$ domain interfaces was calculated with PISA (Krissinel and Henrick, 2007). For multiple sequence alignments of different Ig allotypes, Clustal Omega server (Madeira et al, 2024) and ESPRIPT 3.0 web tool (Robert and Gouet, 2014) were used.

# Data availability

The atomic coordinates and structure factors have been deposited into the Protein Data Bank (PDB), under the accession codes 9QDH (ThomasA$^{323-878}$) and 9QDI (BF3526). The SAXS data of ThomasA have been deposited in the Small Angle Scattering Biological Data Bank (SASBDB) repository, under the accession codes SASDX45 (ThomasA$^{31-1167}$), SASDX55 (ThomasA$^{31-878}$) and SASDX65 (ThomasA$^{323-878}$). Mass spectrometry raw data were deposited on PRIDE (PXD065112) (Perez-Riverol et al, 2025). All

other data are available from the corresponding author upon reasonable request. Source data are provided with this paper.

The source data of this paper are collected in the following database record: biostudies:S-SCDT-10_1038-S44318-025-00518-w.

## Peer review information

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

## Acknowledgements

This work was supported by grants from the "Agencia Estatal de Investigación (AEI) del Ministerio de Ciencia, Innovación y Universidades (MICIU)" (MICIU/AEI/10.13039/501100011033/FEDER, UE) (PID2021-122177NA-I00) to BT and (PID2021-122705OB-I00) to SRC, and the Department of Health, Basque Government (2024333029) to BT. Additionally, MAM-M (JDC2022-048996-I/MRR) and AMG (PRE2022-102570) were supported by MCIN/AEI/10.13039/501100011033 (MAM-M and AMG) and EU "NextGenerationEU/PRTR" (MAM-M). Moreover, BT was supported by "Ramón y Cajal" (RYC2020-028922-I/AEI/ 10.13039/501100011033) and "Programa de movilidad del personal investigador doctor (2023)" (MV_2024_1_0005; Basque Government) fellowships. We acknowledge Diamond Light Source beamlines I24 and B21 (proposal mx34289) and ALBA synchrotron beamline BL13-XALOC (2023087729) for providing access to synchrotron radiation facilities. The authors acknowledge the technical and personnel support provided by the Proteomics and Cell Culture Platforms at IIS Biobizkaia (Spain) for assistance with sample preparation and data analysis. We acknowledge Prof. Gomis-Ruth for the generous donation of IgA2$_{Fab-HR}$ vector.

## Author contributions

**María Ángeles Márquez-Moñino**: Conceptualization; Formal analysis; Investigation; Writing—original draft; Writing—review and editing. **Ana Martínez Gascueña**: Conceptualization; Formal analysis; Investigation; Writing—original draft; Writing—review and editing. **Tala Azzam**: Formal analysis; Investigation; Writing—review and editing. **Andrea Persson**: Formal analysis; Investigation; Writing—review and editing. **Aitor Manzanares-Gomez**: Formal analysis; Investigation. **Marina Aguillo-Urarte**: Formal analysis; Investigation. **Trenton T Brown**: Investigation. **Ainhoa Montero-Sagarminaga**: Investigation. **Rolf Lood**: Formal analysis; Supervision; Writing—review and editing. **Andreas Naegeli**: Formal analysis; Supervision; Writing—review and editing. **Sean R Connell**: Formal analysis; Investigation; Writing—review and editing. **Diego E Sastre**: Formal analysis; Investigation; Writing—review and editing. **Eric J Sundberg**: Formal analysis; Supervision; Writing—review and editing. **Beatriz Trastoy**: Conceptualization; Formal analysis; Supervision; Funding acquisition; Investigation; Writing—original draft; Project administration; Writing—review and editing.

Source data underlying figure panels in this paper may have individual authorship assigned. Where available, figure panel/source data authorship is listed in the following database record: biostudies:S-SCDT-10_1038-S44318-025-00518-w.

## Disclosure and competing interests statement

AP, RL and AN are employees of Genovis A.B., and AN and RL hold shares in the company.

# Expanded View Figures

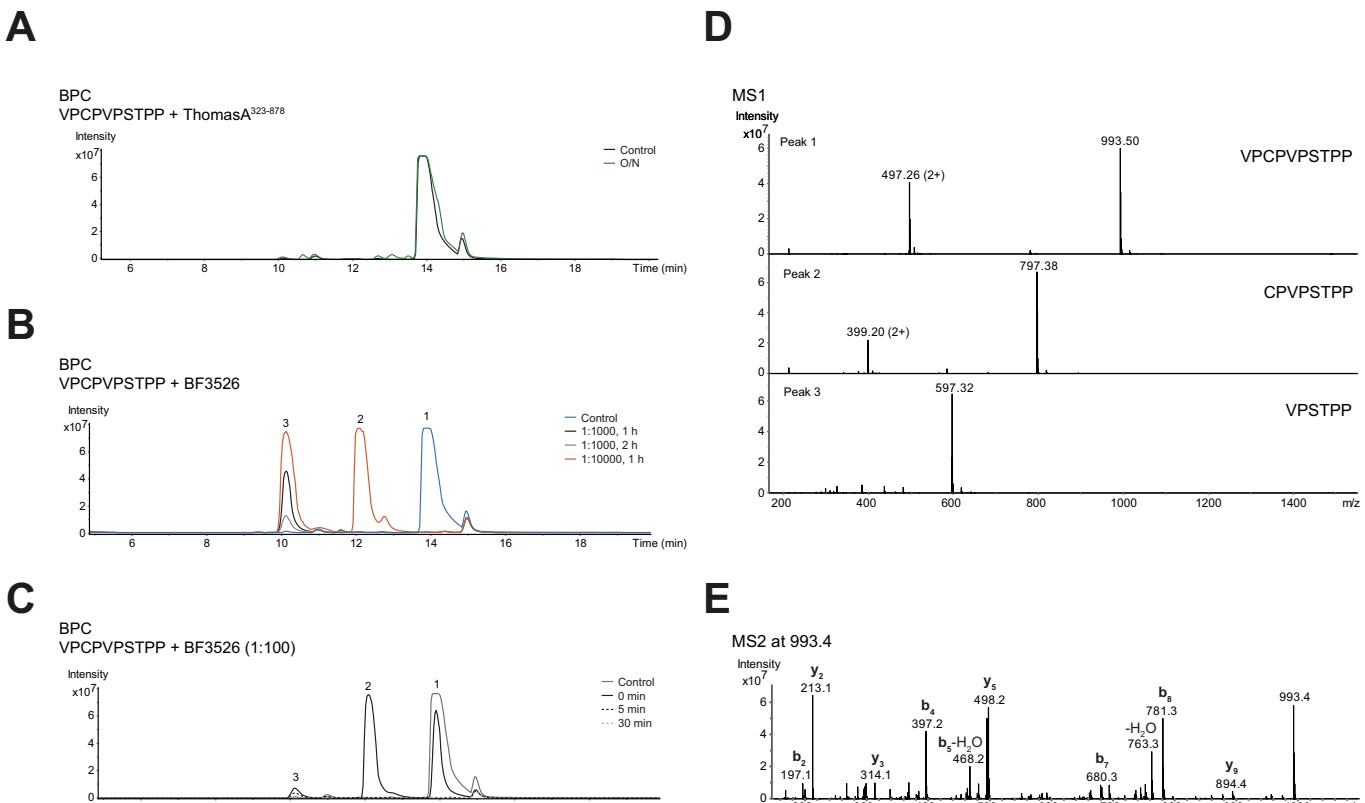

**Figure EV1.  Hydrolytic activity of BF3526 and ThomasA[323-878] against the decapeptide VPCPVPSTPP.**

(A–C) Base peak chromatograms (BPCs) of the reaction of ThomasA[323-878] (A) and BF3526 (B, C) with the decapeptide VPCPVPSTPP. (D) MS1 spectra of Peak 1-3. (E) MS2 at *m/z* 993.4, which corresponds to the intact VPCPVPSTPP peptide. All enzymatic activity measurements were performed in triplicate.

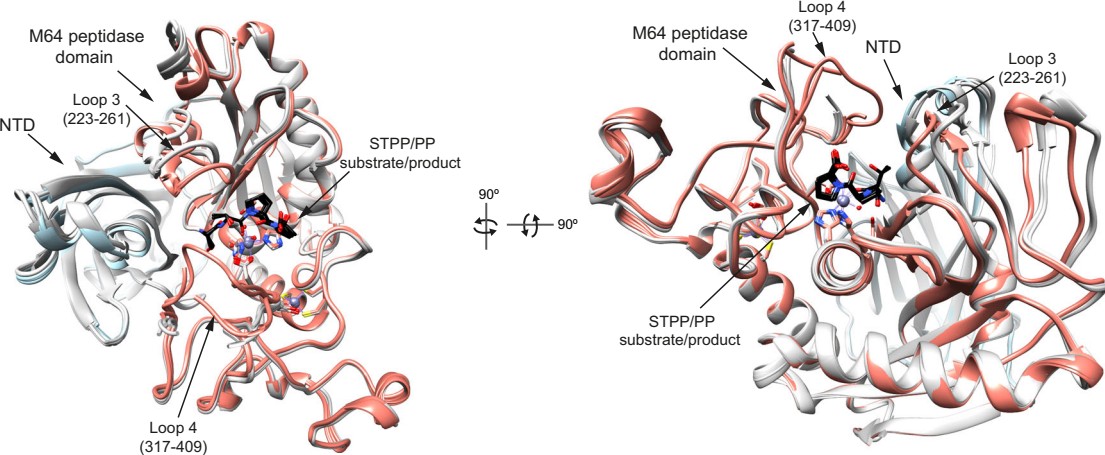

**Figure EV2.  Structural comparison of BF3526-unliganded, BF3526-PP and BF3526-STPP molecules.**

Two views of the superposition of BF3526-unliganded (shown in gray) with BF3526-PP (shown in light blue and salmon) and BF3526-STPP (shown in light blue and salmon).

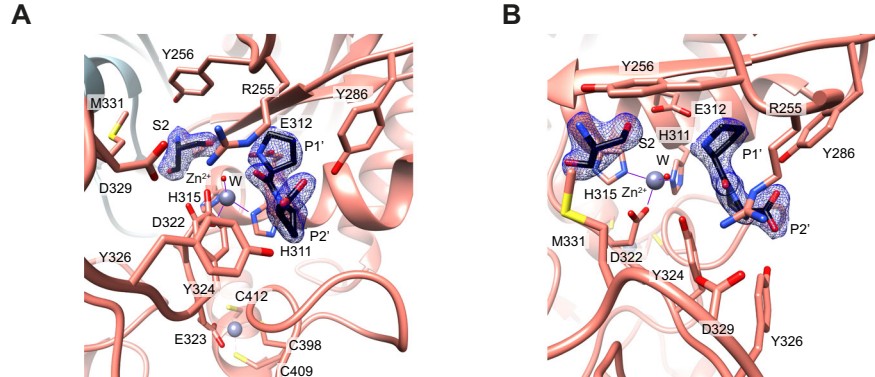

**Figure EV3. Active site of BF3526-PP.**

(A, B) Two views of the BF3526-STPP structure in cartoon representation, showing the location of S2T1P1'P2' substrate within the active site and the key interacting residues. The electron density map of the S2 and P1'P2' product is shown at 1.0 σ r.m.s.d.

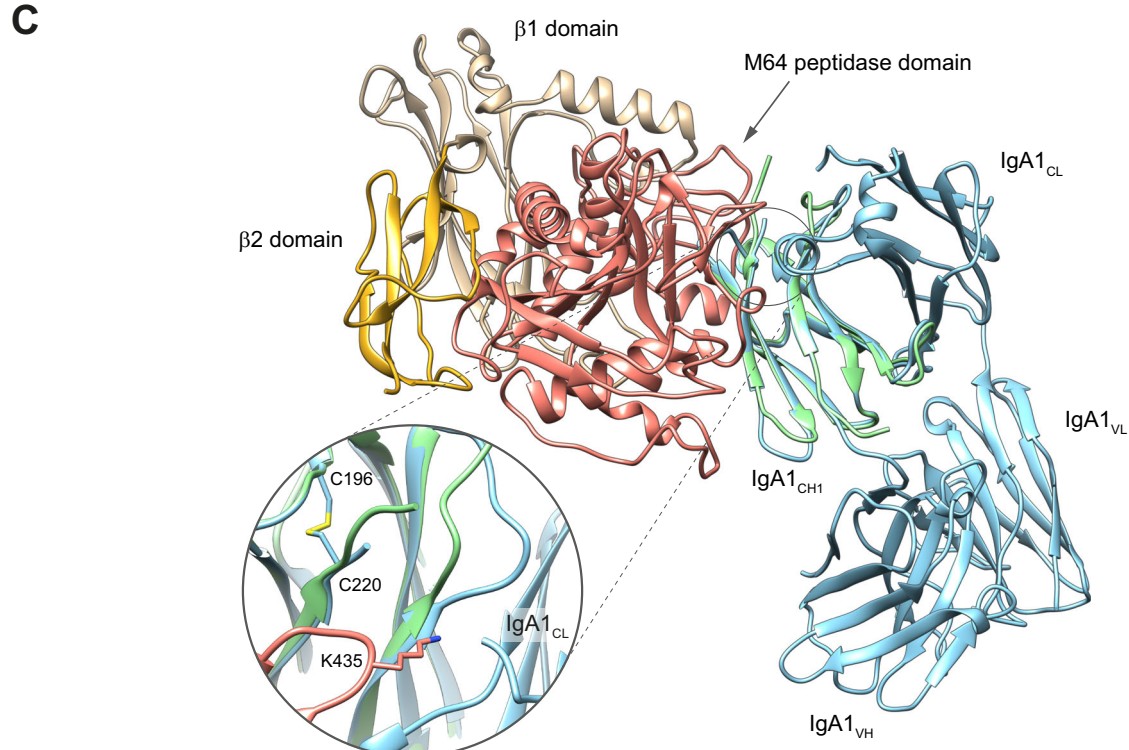

◄  **Figure EV4.  AF3 model analysis of ThomasA$^{323-878}$-IgA1$_{CH1}$ and ThomasA$^{323-878}$-IgA1$_{CH1-CL}$ complexes formation.**

(A, B) Superposition of five ThomasA$^{323-878}$-IgA1$_{CH1}$ (A) and ThomasA$^{323-878}$-IgA1$_{CH1-CL}$ (B) complex models predicted by AF3. M64 peptidase domain, β1, and β2, CH1 and CL domains are colored in salmon, beige, and gold, green and gray, respectively (upper panels). Predicted aligned error (PAE) plots and per-residue confidence scores (pLDDT) for the AF3 complex models are shown in the lower panels. (C) Superposition of IgA1$_{Fab}$ structure (PDB code 7K75, chains A and B, blue) and the AF3 ThomasA$^{323-878}$-IgA1$_{CH1}$ complex model. A close-up view highlights the C196-C220 cysteine bond and the proximity between K435 and the IgA1$_{CL}$ domain. Source data are available online for this figure.

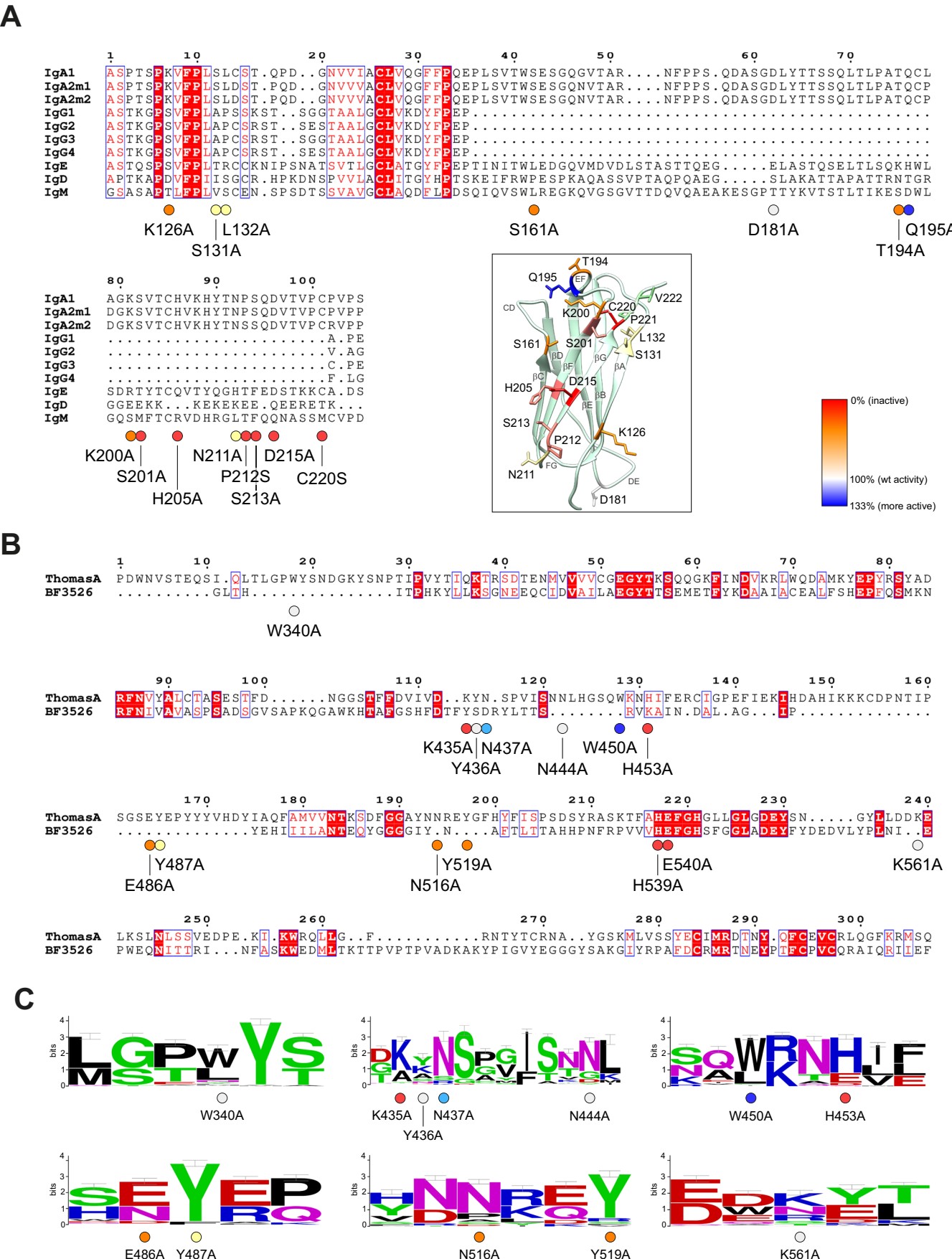

**Figure EV5.  Sequence alignment of ThomasA and BF3526 M64 peptidase domain and CH1 domain of antibody classes.**

(A) Sequence alignment of the CH1 region of IgA1 (UniProt code P01876), IgA2m(1) (UniProt code A0A0G2JMB2), IgA2m(2) (UniProt code P01877), IgG1 (UniProt code P01857), IgG2 (UniProt code P01859), IgG3 (UniProt code P01860), IgG4 (UniProt code P01861), IgE (UniProt code P018541), IgD (UniProt code P01880) and IgM (UniProt code P01871). Inset (black square) shows a cartoon representation of the IgA1$_{CH1}$ domain within the AF3 model of ThomasA$^{323-878}$-IgA1$_{CH1}$ complex. Residues are colored according to the relative activity observed in alanine-scanning experiments. (B) Structural alignment of M64 peptidase domain of ThomasA and BF3526. Mutated residues from alanine-scanning experiments are highlighted with circles, with colors indicating their relative activity on a gradient scale. (C) Multiple sequence alignment of proteins from CL-V of our SSN generated using MUSCLE and WebLogo server. The level of conservation of selected regions that comprise the ThomasA mutants was generated with WebLogo3 (http://weblogo.threeplusone.com).

