## [Peer Review File · The EMBO Journal]

Molecular basis of Fab-dependent IgA antibody recognition by gut-bacterial metallopeptidases

Maria Angeles Marquez-Monino, Ana Martínez Gascueña, Tala Azzam, Andrea Persson, Aitor Manzanares-Gomez, Marina Aguillo-Urarte, Trenton Brown, Ainhoa Montero-Sagarminaga, Rolf Lood, Andreas Naegeli, Sean Connell, Diego Sastre, Eric Sundberg, and Beatriz Trastoy

Corresponding author(s): Beatriz Trastoy (beatriz.trastoybello@bio-bizkaia.eus)

Review Timeline:

Submission Date:	2nd Apr 25
Editorial Decision:	10th May 25
Revision Received:	17th Jun 25
Editorial Decision:	3rd Jul 25
Revision Received:	4th Jul 25
Accepted:	9th Jul 25

Editor: Ioannis Papaioannou

Transaction Report:

Dear Beatriz,

Thank you again for submitting your manuscript EMBOJ-2025-120971 for consideration by The EMBO Journal, and for your patience during peer review. As I have already informed you, your manuscript has now been seen by three experts in the field, and we have received the full set of their detailed, constructive, and well-informed reports, which you can find below.

I am pleased to say that, as you will see, the referees all indicate interest in the study and the findings, recognize the novelty and importance of the work, and conclude that it will make a significant contribution to the field. The reviewers find the work excellent from a biological point of view, but they raise a number of concerns regarding the methods and results of the structural aspects of the study, and they provide detailed suggestions for strengthening the study further, before it can be published in The EMBO Journal.

Given the referees' supportive comments and positive recommendations, I would like to invite you to submit a revised version of your manuscript taking the referees' suggestions on board, along with a detailed point-by-point response addressing all referees' comments. I should add that it is The EMBO Journal policy to allow only a single round of major revision, and acceptance of your manuscript will therefore depend on the completeness of your responses in this revised version. Please let me know if you have any questions or comments that you would like to discuss with me. If there are any major points you do not agree with or cannot address during your revision, I would encourage you to share them with me as early as possible to discuss how to proceed further in the most efficient way.

We generally allow three months as standard revision time (August 9, 2025). As a matter of policy, competing manuscripts published during this period will not negatively impact our assessment of the conceptual advance presented by your study. However, we request that you contact us as soon as possible upon publication of any related work, to discuss how to proceed. Should you foresee a problem in meeting this three-month deadline, please let us know in advance and we will be able to grant an extension.

Thank you for the opportunity to consider your work for publication in The EMBO Journal. I look forward to your revision.

Best wishes,

Ioannis

Instructions for preparing your revised manuscript

1. When you are ready to submit the revision, please upload:

- A Word file of the manuscript text (including legends of main Figures, EV Figures and Tables). Please make sure that changes are highlighted (or "tracked") to be clearly visible.

- Individual production-quality figure files (one file per figure). When assembling your figures, please refer to our figure preparation guidelines in order to ensure proper formatting and readability in print as well as on screen:

If the data shown in a figure are obtained from n {less than or equal to} 2, please use scatter plots showing the individual data points.

i. the name of the statistical test used to generate error bars and P values

ii. the number (n) of independent experiments (please specify technical or biological replicates) underlying each data point

(discussion of statistical methodology can be reported in the Materials and Methods section, but figure legends should contain a basic description of n , P , and the test applied)

iii. the nature of the bars and error bars (s.d., s.e.m.).

- A point-by-point response to the referees' comments, with a detailed description of the changes made (as a word file). All referees' concerns must be fully addressed and their suggestions taken on board. When preparing your letter of response to the referees' comments, please bear in mind that this will form part of the Review Process File and will therefore be available online to the community. Please note that you have the possibility to opt out of the transparent process at any stage prior to publication by letting the editorial office know (contact@embojournal.org); if you do opt out, the Review Process File link will point to the following statement: "No Review Process File is available with this article, as the authors have chosen not to make the review process public in this case.". For more details on our Transparent Editorial Process, please visit our website: <https://www.embopress.org/page/journal/14602075/authorguide#transparentprocess>

- Expanded View (EV) files (replacing Supplementary Information) that are collapsible/expandable online. A maximum of 5 EV Figures can be typeset. EV Figures should be cited as "Figure EV1, Figure EV2" etc. in the text, and their respective legends should be included in the manuscript file after the legends of regular figures. See detailed instructions regarding Expanded View files here: <https://www.embopress.org/page/journal/14602075/authorguide#expandedview>

- For the figures that you do NOT wish to display as Expanded View figures, they should be bundled together with their legends in a single PDF file called "Appendix", which should start with a short Table of Contents (including page numbers). Appendix figures should be referred to in the main text as: "Appendix Figure S1, Appendix Figure S2" etc. Please see detailed instructions here: <https://www.embopress.org/page/journal/14602075/authorguide#expandedview>

- A complete author checklist, which you can download from our author guidelines (<https://www.embopress.org/page/journal/14602075/authorguide>). Please note that the checklist will also be part of the Review Process File.

2. Please note that no statistics should be calculated and shown in Figures if $n=2$. Please also note that each p value should be reported as an exact value.

3. Before submitting your revision, primary datasets (and computer code, where appropriate) produced in this study need to be deposited in appropriate public databases (see <https://www.embopress.org/page/journal/14602075/authorguide#dataavailability>). In particular, you are kindly requested to deposit all structural and mass spectrometry data produced in this study in appropriate databases. The accession numbers, database, and the specific URLs (links) should be listed in a formal "Data availability" section (placed after Methods), following the example below:

"The RNA-seq datasets produced in this study are available in the following database:
Gene Expression Omnibus GSE46843 (<https://www.ncbi.nlm.nih.gov/geo/query/acc.cgi?acc=GSE46843>)"

*** All links should resolve to a page where the data can be accessed. ***

*** Please remember to provide in the Data availability section of your revised manuscript reviewer passwords if the datasets are not yet public. ***

*** The Data Availability Section is restricted to new primary data that are part of this study. In case you have no data that require deposition in a public database, please state so instead of referring to the database: "Our study includes no data deposited in public repositories." under the heading "Data availability". ***

4. The materials and methods need to be described in the manuscript using our structured methods format, which is now required for all research articles. According to this format, the Methods section includes a single "Reagents and Tools Table" - listing key reagents, experimental models, software and relevant equipment including their sources and relevant identifiers - followed by a "Methods and Protocols" section describing the methods. Please download and fill our Reagents and Tools Table template (.docx), which you can find in our author guide:

<https://www.embopress.org/page/journal/14602075/authorguide#structuredmethods>. When submitting your revised manuscript, please do not include the Reagents and Tools Table in the Methods section of the manuscript but instead upload it as a separate file choosing the file type "Reagent Table".

5. Please check that the title and the abstract of the manuscript are brief, yet explicit, even to non-specialists. The length of the title should not exceed 100 characters, and the abstract should be a single paragraph not exceeding 175 words.

6. Please also note our reference format: <https://www.embopress.org/page/journal/14602075/authorguide#referencesformat>.

8. Please remember: digital image enhancement is acceptable practice, as long as it accurately represents the original data and conforms to community standards. If a figure has been subjected to significant electronic manipulation, this must be noted in the

figure legend or in the "Materials and Methods" section. The editors reserve the right to request original versions of figures and the original images that were used to assemble the figure.

9. Our journal encourages inclusion of data citations in the reference list to directly cite datasets that were obtained from public databases. Data citations in the article text are distinct from normal bibliographical citations and should directly link to the database records from which the data can be accessed. In the main text, data citations are formatted as follows: "Data ref: Smith et al, 2001" or "Data ref: NCBI Sequence Read Archive PRJNA342805, 2017". In the Reference list, data citations must be labeled with "[DATASET]". A data reference must provide the database name, accession number/identifiers, and a resolvable link to the landing page from which the data can be accessed at the end of the reference. Further instructions are available at: <https://www.embopress.org/page/journal/14602075/authorguide#referencesformat>.

10. We request authors to consider both actual and perceived competing interests. Please review our policy (<https://www.embopress.org/page/journal/14602075/authorguide#conflictsofinterest>) and update your competing interests statement if necessary. Please name this section 'Disclosure and competing interests statement' and place it after the Acknowledgements section.

11. Please note that all corresponding authors are required to provide an ORCID ID upon submission of a revised manuscript (<https://orcid.org/>). Please find instructions on how to link your ORCID ID to your account in our manuscript tracking system in our Author guidelines (<https://www.embopress.org/page/journal/14602075/authorguide#authorshipguidelines>).

12. We use CRediT to specify the contributions of each author in the journal submission system. CRediT replaces the author contribution section, which should be removed from the manuscript. Please use the free text box to provide more detailed descriptions. See also guide to authors: <https://www.embopress.org/page/journal/14602075/authorguide#authorshipguidelines>.

14. We would also welcome the submission of cover suggestions or motifs to be used by our Graphics Illustrator in designing a cover.

15. Please use the link below to submit your revision:

XXXXXXXXXXXXXXXXXXXXXXXXXXXX

Referee #1:

Márquez-Moñino et al. present a comprehensive and well-executed study detailing the catalytic and molecular mechanisms of substrate recognition by the M64 metallopeptidases ThomasA from *T. ramosa* and BF3526 from *B. fragilis*. Their work demonstrates that these peptidases exhibit distinct substrate specificities: ThomasA cleaves IgA through exclusive recognition of the Fab region, a mechanism that differs from other antibody-specific peptidases, whereas BF3526 targets the N-terminus of predigested proteins. The integration of structural, biochemical, and mutagenesis data provides strong and convincing support for the authors' conclusions.

Overall, the manuscript is clearly written and represents a significant and valuable contribution to the field. I recommend publication in *The EMBO Journal* after minor revisions addressing the points listed below.

General remarks:

1. Regarding the SAXS data, Supplementary Table 2 shows an 18% difference between the experimental and theoretical molecular weight of the ThomasA31-1167 construct, which exceeds typical experimental error. The authors should clarify whether this discrepancy is due to sample aggregation and whether it is observed in all samples analyzed. It is also suggested to include a graph of the SEC-SAXS elution profiles in the supplementary figures, indicating the selected frame range, to demonstrate the absence of contamination by larger oligomers.

2. In Figure 2, the authors present the superposition of the models generated with AF3 for the analyzed ThomasA constructs. However, they have not included an analysis using CRYSQL to compare the AlphaFold models with the experimental SAXS data. Including such an analysis would be appropriate to assess the accuracy of the models. It is also recommended to include the PAE plots and pLDDT scores of the AF3 models in the supplementary figures and to provide the AF3 model coordinate files as Source Data.

3. The authors show that the AF3 model of ThomasA31-1167 does not fit the SAXS envelope, suggesting a different positioning of the β 3- β 6 domains. Have the authors considered using programs that refine multiple conformations of the AF model and fit

them to the SAXS data, such as MultiFoXS?

4. In Supplementary Figure 6, the chi-squared value of 5.4 obtained with CRYSOLOG indicates a moderate to poor fit between the crystal structure of ThomasA323-878 and the experimental SAXS data. The authors should discuss potential explanations for this discrepancy.
5. It is recommended that the SAXS data be deposited in the SASBDB repository to enhance data transparency and reproducibility.
6. In the BF3526 crystal structure, although the text states that no major conformational changes occur between the ligand-free and dipeptide-bound forms of BF3526, a movement in loop 3 is clearly observed (Supplementary Figure 13). The authors should clarify this point.
7. The hydrolytic activity values presented in Figure 4A and Figure 4E are inconsistent. For example, the relative Fab production ratio of ThomasA31-1167 at 45 minutes is approximately 0.8 in panel A and about 0.4 in panel E. Were these values obtained under different experimental conditions? Clarification is needed.
8. Could the authors discuss why the smaller ThomasA construct cleaves the IgA2Fab-HR construct with similar efficiency to IgA1, whereas in other constructs the cleavage efficiency is reduced?
9. The authors performed an alanine-scanning mutagenesis based on the ThomasA-IgA1 AF3 model to identify key residues for catalytic activity. Have they verified that these mutations do not affect the overall structure or stability of the mutated proteins? Additionally, for the mutations that increased hydrolytic activity, could the authors propose a molecular mechanism explaining this phenotype?
10. The number of biological or technical replicates should be indicated to ensure reproducibility and transparency (Figures 4E, 9A, 9B).

Other specific suggestions:

1. Line 98: A full stop is missing at the end of the sentence.
2. Line 150: "8qk8" should be written in uppercase ("8QK8") to be consistent with the nomenclature used throughout the manuscript.
3. Line 152: Similarly, "7yl4" should be corrected to uppercase ("7YL4").
4. Line 162: The SAXS data are only presented in Supplementary Table 2; therefore, the reference to Supplementary Table 1 should be removed to avoid confusion.
5. Line 166: Supplementary Table 1 does not include information regarding protein expression of the constructs, contrary to what is implied in the text.
6. Line 167: It should be clearly stated that the molecular weights were calculated from experimental SAXS data. It is currently unclear whether the values were derived from gel filtration chromatography elution volumes.
7. Lines 451-452: "H435" should be corrected to "H453" to reflect the correct residue number.
8. Line 642: "Es.coli" should be corrected to "E. coli".
9. Line 734: "0.25 ml" should be corrected to "0.25 µl" as this appears to be a typographical error.
10. Line 1167, Figure 5 legend: The labels "a" and "b" should be capitalized ("A" and "B") for consistency.
11. Figure 8C: The figure legend should include an explanation of the color coding used in the panel.
12. Supplementary Figure 2: Remove the extra parenthesis before "BACOVA...".
13. Supplementary Figure 2: It is recommended that the authors specify which chain was used for the structural superposition (e.g. structure 4DF9 has six chains in the asymmetric unit).
14. Supplementary Figure 2: In the expression "(c). d", the lowercase "c" and "d" should be capitalized.
15. Supplementary Figure 4: The phrase "and BF3526 d for x-ray crystallography" should be corrected to "and BF3526 used for x-ray crystallography".
16. Supplementary Figure 4: The reference to "rods (E)" should be corrected to "rods (F)".
17. Supplementary Figure 6: The software name "CRYSOLOG" should be written entirely in uppercase letters.
18. Supplementary Figure 6B: Provide two different views to better illustrate the SAXS fit.
19. Supplementary Figure 10: The labels "a" and "b" should be capitalized ("A" and "B") in the figure to be consistent with the legend.

Referee #2:

- general summary and opinion about the principal significance of the study, its questions and findings
- Several commensal and pathogenic bacteria can colonise and persist in mucosal tissue by exploiting proteases able to cleave and inactivate IgA antibodies; IgA are the most abundant antibodies in mucosal tissue. These IgA-specific proteases (IgAPs) gained enormous attention recently due to their dual pharmaceutical role as antibiotic targets and as potential therapeutics of IgA nephropathy, a currently incurable autoimmune disease, which is believed to be caused by unusual glycosylation patterns on IgA antibodies.
- Several of these IgA-specific proteases (IgAPs) were characterised, including M26 (metalloprotease) and S6 (serine protease) families. These proteases cleave in the hinge region of the IgA, connecting Fc and Fab segments. The authors analyse two members from another metalloprotease subfamily, M64, which are able to cleave the IgA1 and IgA2 isoforms: ThomasA (from *Thomasclavelia ramosa*) and BF3526 from *Bacteroides fragilis*. The authors investigated the relevance of the individual domains

within ThomasA and the basis for the IgA specificity by testing relevant fusion constructs, different substrates with a broad method spectrum. Overall, this is a very important work.

- specific major concerns essential to be addressed to support the conclusions
 - a) Line 240ff: "None of the ThomasA constructs were able to hydrolyze the IgA1 HR within GST-IgA1HR-GH18L over time, suggesting that the enzyme requires the Fab and/or Fc regions of IgA1 for recognition and interaction with the substrate (Supplementary Fig. 8)". What worries me are the newly introduced fusion domains. Specifically, the GST fusion will generate a dimeric protein, which is likely to affect the accessibility of the protease. Why not using smaller fusion tags like His-tag? Or a monomeric fusion tag like MBP.
 - b) Supplementary table 3: Rmerge, Rmeas values of ThomasA are high; test the merging statistics for each of the three 2-fold axis separately, to check / exclude whether the space group may have lower symmetry (e.g. monoclinic)
 - c) Line 341ff "The STPP substrate and PP product were unambiguously modeled in the active site of the enzyme, occupying equivalent positions within the narrow cavity flanked by $\alpha 4$, $\beta 12$, and loops 3, 4 and NTD (Fig. 6 and 7 and Supplementary Fig. 13)." The electron density shown in figure 7 A for the substrate ST-PP indicates that the substrate was co-crystallized. Why is the peptide not hydrolysed? The supplement should show the complex together with the electron density (1) in stereo and, of course, (2) in standard orientation. Is the catalytic water present? Is the T-P peptide bond in trans, or in cis?

- minor concerns that should be addressed
 - a) Suppl. Fig. 12: While the electron density is shown in stereo, the cartoon is apparently broken. This is most evident in the orientation of the terminal helix which differs in the two stereo figures by approximately 30{degree sign} rather than 3{degree sign}. Please correct.
 - b) Figure 6, 7, Suppl. Fig. 15: B-D: non-standard orientation. What is standard orientation? Look at suppl. figure 15 A. When active sites are presented in standard orientation, the substrate runs from N- to C-terminus from left to right. Following this convention is very helpful for understanding the always challenging structural representations, and in a journal with a broad readership such as EMBO J, I consider it a must. Please correct this view, it is really easy, it essentially means to rotate the figure for 180{degree sign} around z, i.e. the axis perpendicular to the paper plane. IGNORing this convention is, well, you can guess it ;-)
 - c) "Fab region and IgA1 HR fused to the Fc region of IgG1" - is NOT (or only partly shown in figure 4D).
 - d) Line 250. While I like the choice of the IgA-IgG fusion constructs (of which only one expressed), I am sceptical with interpretation that the HR domain alone is insufficient for substrate recognition. I find it very optimistic to assume that the fused domains would behave neutral. Remember Hippocrates: first, do no harm!

- any additional non-essential suggestions for improving the study (which will be at the author's/editor's discretion)

Frankly, I found the manuscript rather difficult to read. Maybe the structure of the paper can be made clearer.

Referee #3:

I apologize for the tardy submission of the report on this manuscript. I read it back and forth carefully to find any points I could criticize, but I found nothing. Since I'm not an expert in structural biology, I concentrated on experimental design and how the research was executed. From a biological point of view, the manuscript is perfect, and the conclusions are based on results generated by complementary methods such as AF3 modeling, SAXS, and X-ray crystallography. Along with the results of mutagenetic experiments, the manuscript presents an unbiased molecular model of the structure and function of IgA proteases belonging to the M64 family of metalloproteases. The divergent substrate recognition strategies used by two representative proteases are fascinating in the context of the conserved catalytic mechanism. In addition, the paper is very well written and illustrated; it is a true pleasure to read. I hope other reviewers also find this manuscript presents an outstanding piece of work deserving of publication in EMBO J.

POINT BY POINT RESPONSE

Referee #1:

Márquez-Moñino et al. present a comprehensive and well-executed study detailing the catalytic and molecular mechanisms of substrate recognition by the M64 metallopeptidases ThomasA from *T. ramosa* and BF3526 from *B. fragilis*. Their work demonstrates that these peptidases exhibit distinct substrate specificities: ThomasA cleaves IgA through exclusive recognition of the Fab region, a mechanism that differs from other antibody-specific peptidases, whereas BF3526 targets the N-terminus of predigested proteins. The integration of structural, biochemical, and mutagenesis data provides strong and convincing support for the authors' conclusions.

Overall, the manuscript is clearly written and represents a significant and valuable contribution to the field. I recommend publication in The EMBO Journal after minor revisions addressing the points listed below.

General remarks:

ANSWER: We thank the reviewer for all the constructive comments/suggestions to improve the manuscript.

1. Regarding the SAXS data, Supplementary Table 2 shows an 18% difference between the experimental and theoretical molecular weight of the ThomasA³¹⁻¹¹⁶⁷ construct, which exceeds typical experimental error. The authors should clarify whether this discrepancy is due to sample aggregation and whether it is observed in all samples analyzed.

ANSWER: We thank the reviewer for bringing this to our attention. We have verified that the observed difference between the experimental and theoretical molecular weight of the ThomasA³¹⁻¹¹⁶⁷ construct is not due to sample aggregation or oligomerization. We believe this deviation is caused by the inherent flexibility and elongated shape of the ThomasA³¹⁻¹¹⁶⁷ construct, which can affect the accuracy of molecular weight estimation by SAXS (Rambo & Tainer, 2013). Importantly, this discrepancy was not consistently observed across all constructs analyzed, suggesting that it is specific to the structural features of this construct.

2. It is also suggested to include a graph of the SEC-SAXS elution profiles in the supplementary figures, indicating the selected frame range, to demonstrate the absence of contamination by larger oligomers.

ANSWER: We thank the reviewer for this suggestion. The signal plot of each SEC-SAXS experiment is represented in Figure R1, with the selected frame ranges clearly indicated. These ranges correspond to well-defined peak regions with low variation in R_g , supporting the absence of contamination by larger oligomeric species.

Figure R1. | SEC-SAXS Signal Plot of ThomasA constructs. A-C SEC-SAXS Signal Plot (gray) of ThomasA³¹⁻¹¹⁶⁷ (A), ThomasA³¹⁻⁸⁷⁸ (B) and ThomasA³²³⁻⁸⁷⁸ (C). The selected range of frames used for processing each dataset have been highlighted in a colored region.

3. In Figure 2, the authors present the superposition of the models generated with AF3 for the analyzed ThomasA constructs. However, they have not included an analysis using CRY SOL to compare the AlphaFold models with the experimental SAXS data. Including such an analysis would be appropriate to assess the accuracy of the models.

ANSWER: We have included a new Appendix Figure 3 presenting the CRY SOL analysis comparing the experimental SAXS data of the three ThomasA constructs with the theoretical scattering curves calculated from their respective AF3 models. This addition provides a direct assessment of the agreement between the predicted structures and the solution data.

4. It is also recommended to include the PAE plots and pLDDT scores of the AF3 models in the supplementary figures and to provide the AF3 model coordinate files as Source Data.

ANSWER: We have included the PAE plots and pLDDT scores of the AF3 models in the new Appendix Figure S3. We have also included the AF3 model coordinates used in this study as Source Data.

5. The authors show that the AF3 model of ThomasA³¹⁻¹¹⁶⁷ does not fit the SAXS envelope, suggesting a different positioning of the β 3- β 6 domains. Have the authors considered using programs that refine multiple conformations of the AF model and fit them to the SAXS data, such as MultiFoXS?

ANSWER: We did explore the use of ensemble modeling approaches to improve the fit of the AF3 model of ThomasA³¹⁻¹¹⁶⁷ to the SAXS data, including the use of CORAL, Robetta or MultiFoXS. However, these attempts did not result in a significantly better fit, suggesting that the conformational differences observed, particularly in the positioning of the β 3- β 6 domains, may reflect structural variability not easily captured by the current AF3 model or by flexible fitting alone.

6. In Supplementary Figure 6, the chi-squared value of 5.4 obtained with CRY SOL indicates a moderate to poor fit between the crystal structure of ThomasA³²³⁻⁸⁷⁸ and the experimental SAXS data. The authors should discuss potential explanations for this discrepancy.

ANSWER: The χ^2 value of 5.4 obtained with CRY SOL indeed indicates a moderate fit between the crystal structure of ThomasA³²³⁻⁸⁷⁸ and the experimental SAXS data. While we were unable to improve this value through further data processing, one plausible explanation is that the protein exhibits a degree of conformational flexibility in solution that is not captured in the static crystal structure.

7. It is recommended that the SAXS data be deposited in the SASBDB repository to enhance data transparency and reproducibility.

ANSWER: We thank the reviewer for bringing this to our attention. The SAXS data of ThomasA³¹⁻¹¹⁶⁷, ThomasA³¹⁻⁸⁷⁸ and ThomasA³²³⁻⁸⁷⁸ have been deposited in the SASBDB data repository under the accession codes SASDX45, SASDX55 and SASDX65, respectively.

8. In the BF3526 crystal structure, although the text states that no major conformational changes occur between the ligand-free and dipeptide-bound forms of BF3526, a movement

in loop 3 is clearly observed (Supplementary Figure 13). The authors should clarify this point.

ANSWER: We have clarified the text regarding this comment: *“No major conformational changes are observed across the three enzyme states, except for connecting loop β 10- α 3 (loop 3, residues 223-261), which shifts closer to the active site in the peptide-bound structures, and connecting loop α 4- α 5 (loop 4, residues 317-409), which is only structured in chain C of the BF3526-PP complex, suggesting flexibility above the active site”*

9. The hydrolytic activity values presented in Figure 4A and Figure 4E are inconsistent. For example, the relative Fab production ratio of ThomasA31-1167 at 45 minutes is approximately 0.8 in panel A and about 0.4 in panel E. Were these values obtained under different experimental conditions? Clarification is needed

ANSWER: We thank the reviewer for bringing this to our attention. The apparent discrepancy between the hydrolytic activity values in Figures 4A and 4E (now 3A and 3E) arises because the two panels represent different experimental setups. Figure 3A shows a time-course digestion of full-length IgA, where aliquots were directly sampled approximately every 6.5 minutes and immediately analyzed by LC-MS. In contrast, Figure 3E presents endpoint measurements from separate cleavage reactions that were stopped after 45 minutes by the addition of 10 μ L of a mixture of 15 mM EDTA and 0.07% formic and then analyzed by LC-MS. We have clarified these procedural details in the Methods section.

10. Could the authors discuss why the smaller ThomasA construct cleaves the IgA2Fab-HR construct with similar efficiency to IgA1, whereas in other constructs the cleavage efficiency is reduced?

ANSWER: We thank the reviewer for pointing this out. Upon reprocessing the data, we found that the mass peaks corresponding to the IgA2_{Fab-HR} substrate and product had been incorrectly assigned. We have corrected this in the revised analysis and updated Figure 3 accordingly. We can now confirm that all three ThomasA constructs show reduced activity against the IgA2_{Fab-HR}.

11. The authors performed an alanine-scanning mutagenesis based on the ThomasA-IgA1 AF3 model to identify key residues for catalytic activity. Have they verified that these mutations do not affect the overall structure or stability of the mutated proteins?

ANSWER: We thank the reviewer for this important point. Both wild-type and mutant ThomasA and IgA proteins were analyzed by size exclusion chromatography (SEC). All mutant samples showed a principal elution peak that overlapped with their respective wild-type counterparts, as shown in Figure R2 below. This suggests that the mutations did not result in significant overall structural changes or compromise protein stability.

Figure R2. Size exclusion chromatography (SEC) profiles of ThomasA³²³⁻⁸⁷⁸ and IgA1_{fl} mutants. A-B SEC profiles of wild-type and mutant constructs of ThomasA³²³⁻⁸⁷⁸ (A) and IgA1_{fl} (B), obtained using a Superdex 200 column HiLoad 10/300GL. Wild-type proteins are shown in black; the different mutants are colored as indicated in the legend.

12. Additionally, for the mutations that increased hydrolytic activity, could the authors propose a molecular mechanism explaining this phenotype?

ANSWER: We thank the reviewer for this valuable observation. Although we do not have experimental data directly explaining the increased activity of the ThomasA-N437A and W450A variants or the improved substrate efficiency of IgA1-Q195A, we propose a tentative interpretation based on the AlphaFold model and mutagenesis data (Figure 9). The moderate increases in activity (~23%) may result from local changes that subtly favor substrate binding or access to the active site. For example, W450 is located in a polar-rich region near IgA1 residues important for activity, and its replacement might facilitate polar interactions at the interface. N437 lies near the active site and may partially obstruct access, so its removal could improve substrate accommodation. Q195 from IgA1 is positioned near a predicted salt bridge and hydrophobic residues; its substitution might locally stabilize the interface by reducing polarity.

13. The number of biological or technical replicates should be indicated to ensure reproducibility and transparency (Figures 4E, 9A, 9B).

ANSWER: We thank the reviewer for highlighting the importance of transparency regarding replicates. We have now indicated the number of biological and/or technical replicates in the figures to ensure clarity and reproducibility.

Other specific suggestions:

14. Line 98: A full stop is missing at the end of the sentence.

ANSWER: We have corrected this in the main text.

15. Line 150: "8qk8" should be written in uppercase ("8QK8") to be consistent with the nomenclature used throughout the manuscript.

ANSWER: We have corrected this in the main text.

16. Line 152: Similarly, "7yl4" should be corrected to uppercase ("7YL4").

ANSWER: We have corrected this in the main text.

17. Line 162: The SAXS data are only presented in Supplementary Table 2; therefore, the reference to Supplementary Table 1 should be removed to avoid confusion.

ANSWER: We thank the reviewer for the observation. While the SAXS data itself is indeed presented in Supplementary Table 2 (now Appendix Table 2), Supplementary Table 1 (now Appendix Table 1) provides essential information on the constructs and primers used in the SAXS experiments. To maintain clarity and ensure completeness, we believe it is appropriate to retain references to both Appendix Tables 1 and 2 in the text.

18. Line 166: Supplementary Table 1 does not include information regarding protein expression of the constructs, contrary to what is implied in the text.

ANSWER: We agree with the reviewer. We have removed the reference to Supplementary Table 1 in the text.

19. Line 167: It should be clearly stated that the molecular weights were calculated from experimental SAXS data. It is currently unclear whether the values were derived from gel filtration chromatography elution volumes.

ANSWER: We have added a comment in the main text: *"The three successfully expressed constructs eluted from the size exclusion chromatography column as monomers, with average molecular weights of 155.1, 91.7, and 69.2 kDa, respectively, calculated from experimental SAXS data (Appendix Table S2)."*

20. Lines 451-452: "H435" should be corrected to "H453" to reflect the correct residue number.

ANSWER: We have corrected this in the main text.

21. Line 642: "Es.coli" should be corrected to "E. coli".

ANSWER: We have corrected this in the main text.

22. Line 734: "0.25 ml" should be corrected to "0.25 μ l" as this appears to be a typographical error.

ANSWER: We have corrected this in the main text.

23. Line 1167, Figure 5 legend: The labels "a" and "b" should be capitalized ("A" and "B") for consistency.

ANSWER: We have corrected this in the main text.

24. Figure 8C: The figure legend should include an explanation of the color coding used in the panel.

ANSWER: We have corrected this in the main text.

25. Supplementary Figure 2: Remove the extra parenthesis before "BACOVA...".

ANSWER: We have corrected this in the Appendix.

26. Supplementary Figure 2: It is recommended that the authors specify which chain was used for the structural superposition (e.g. structure 4DF9 has six chains in the asymmetric unit).

ANSWER: We have corrected this in the figure legend.

27. Supplementary Figure 2: In the expression "(c). d", the lowercase "c" and "d" should be capitalized.

ANSWER: We have corrected this in the Appendix.

28. Supplementary Figure 4: The phrase "and BF3526 d for x-ray crystallography" should be corrected to "and BF3526 used for x-ray crystallography".

ANSWER: We have corrected this in the Appendix.

29. Supplementary Figure 4: The reference to "rods (E)" should be corrected to "rods (F)".

ANSWER: We have corrected this in the Appendix.

30. Supplementary Figure 6: The software name "CRY SOL" should be written entirely in uppercase letters.

ANSWER: We have corrected this in the Appendix.

31. Supplementary Figure 6B: Provide two different views to better illustrate the SAXS fit.

ANSWER: We have modified this figure, now Appendix Figure S9B.

32. Supplementary Figure 10: The labels "a" and "b" should be capitalized ("A" and "B") in the figure to be consistent with the legend.

ANSWER: We have modified this figure, now Appendix Figure S11.

Referee #2:

- general summary and opinion about the principal significance of the study, its questions and findings

33. Several commensal and pathogenic bacteria can colonise and persist in mucosal tissue by exploiting proteases able to cleave and inactivate IgA antibodies; IgA are the most abundant antibodies in mucosal tissue. These IgA-specific proteases (IgAPs) gained enormous attention recently due to their dual pharmaceutical role as antibiotic targets and as potential therapeutics of IgA nephropathy, a currently incurable autoimmune disease, which is believed to be caused by unusual glycosylation patterns on IgA antibodies. Several of these IgA-specific proteases (IgAPs) were characterised, including M26 (metalloprotease) and S6 (serine protease) families. These proteases cleave in the hinge region of the IgA, connecting Fc and Fab segments. The authors analyse two members from another metalloprotease subfamily, M64, which are able to cleave the IgA1 and IgA2 isoforms: ThomasA (from *Thomasclavelia ramosa*) and BF3526 from *Bacteroides fragilis*. The authors investigated the relevance of the individual domains within ThomasA and the basis for the IgA specificity by testing relevant fusion constructs, different substrates with a broad method spectrum. Overall, this is a very important work.

ANSWER: We thank the reviewer for all the constructive comments/suggestions in order to improve the manuscript. We appreciate the reviewer's positive comment on the overall importance of the work.

- specific major concerns essential to be addressed to support the conclusions

34. Line 240ff: "None of the ThomasA constructs were able to hydrolyze the IgA1 HR within GST-IgA1HR-GH18L over time, suggesting that the enzyme requires the Fab and/or Fc regions of IgA1 for recognition and interaction with the substrate (Supplementary Fig. 8)". What worries me are the newly introduced fusion domains. Specifically, the GST fusion will

generate a dimeric protein, which is likely to affect the accessibility of the protease. Why not using smaller fusion tags like His-tag? Or a monomeric fusion tag like MBP.

ANSWER: To address the concern about dimerization of the GST fusion construct, we generated an alternative substrate using a monomeric fusion tag. Specifically, we cloned, expressed, and purified an MBP-TEV-IgA1_{HR}-EGFP construct, in which the IgA1 hinge region was inserted immediately after the TEV cleavage site in the pMLG32 MBP-TEV-EGFP vector. We then performed SDS-PAGE analysis to assess the hydrolytic activity of all three ThomasA constructs against this monomeric MBP-based substrate. These results are now included in Appendix Figure S6A,B.

None of the ThomasA variants were able to hydrolyze the MBP-TEV-IgA1_{HR}-EGFP construct, while TEV protease successfully cleaved its canonical recognition site upstream of the hinge region. This confirms that the IgA1 hinge region is accessible in the MBP-fusion context and that lack of ThomasA activity is not due to tag-induced steric hindrance. Although EGFP can weakly dimerize at high concentrations (>5 mg mL⁻¹) (Snapp *et al*, 2003), we confirmed the oligomeric state of MBP-TEV-IgA1_{HR}-EGFP by size-exclusion chromatography at 0.5 mg mL⁻¹ (Appendix Figure S6F). Importantly, the hydrolytic assays were performed at an even lower concentration (0.1 mg mL⁻¹), further minimizing any potential dimerization effects.

Taken together with the lack of activity against IgG1_{Fab}-IgA1_{HR-Fc} and a synthetic decapeptide corresponding to the IgA hinge region, these results strongly support the conclusion that the IgA1 hinge region alone is insufficient for substrate recognition by ThomasA.

35. Supplementary table 3: R_{merge} , R_{meas} values of ThomasA are high; test the merging statistics for each of the three 2-fold axis separately, to check / exclude whether the space group may have lower symmetry (e.g. monoclinic)

ANSWER: We reanalyzed the data using AIMLESS (CCP4) and assessed the merging statistics for each of the three 2-fold axes separately. The R_{meas} values were found to be equivalent across all three axes, indicating no signs of lower symmetry, such as a monoclinic space group.

The overall R_{merge} and R_{meas} values for the ThomasA dataset are indeed high, but this can be attributed to the high multiplicity of the dataset. This is supported by the comparatively low R_{pim} values, which more accurately reflect data quality under these conditions.

36. Line 341ff "The STPP substrate and PP product were unambiguously modeled in the active site of the enzyme, occupying equivalent positions within the narrow cavity flanked by α 4, β 12, and loops 3, 4 and NTD (Fig. 6 and 7 and Supplementary Fig. 13)." The electron density shown in figure 7 A for the substrate ST-PP indicates that the substrate was co-crystallized. Why is the peptide not hydrolysed? The supplement should show the complex together with the electron density (1) in stereo and, of course, (2) in standard orientation. Is the catalytic water present? Is the T-P peptide bond in trans, or in cis?

ANSWER: We appreciate the reviewer's insightful comment. We have included a new Appendix Figure S13 to show the electron density map in stereo and standard orientation of (1) BF3526-unliganded, (2) BF3526-PP complex and (3) BF3526-STPP complex. We were also intrigued to observe clear electron density for the STPP substrate in two molecules of

the active enzyme in the asymmetric unit, despite the co-crystallization conditions. Our interpretation is that STPP is a low-affinity substrate, likely due to the enzyme's intrinsic preference for cleaving peptide bonds after proline. This may lead to partial or slow hydrolysis and entrapment of the intact peptide in the active site during crystallization.

Importantly, in the two molecules where the STPP peptide was modeled, no clear electron density was observed for the catalytic water molecule, suggesting that the hydrolysis step may not have occurred or was not complete under the crystallization conditions (see new Appendix Figures S13 and S15). In the refined BF3526-STPP complex structure, we did observe additional electron density in the active site. However, this density did not correspond to a water molecule and may be attributable to slight movement or alternative conformations of the carbonyl group of the substrate (Appendix Fig. S15). The omit FEM map generated by removing STPP from the active site supports this interpretation, indicating that the predominant species in the active site adopts this conformation, rather than representing another component of the reaction (Appendix Fig. S15).

Regarding the stereochemistry of the T-P peptide bond, in all modeled cases, the bond adopts a trans conformation, consistent with the expected geometry in the enzyme-substrate complex.

- minor concerns that should be addressed

37. Suppl. Fig. 12: While the electron density is shown in stereo, the cartoon is apparently broken. This is most evident in the orientation of the terminal helix which differs in the two stereo figures by approximately 30{degree sign} rather than 3{degree sign}. Please correct.

ANSWER: We have modified this figure, now Appendix Figure S12.

38. Figure 6, 7, Suppl. Fig. 15: B-D: non-standard orientation. What is standard orientation? Look at suppl. figure 15 A. When active sites are presented in standard orientation, the substrate runs from N- to C-terminus from left to right. Following this convention is very helpful for understanding the always challenging structural representations, and in a journal with a broad readership such as EMBO J, I consider it a must. Please correct this view, it is really easy, it essentially means to rotate the figure for 180{degree sign} around z, i.e. the axis perpendicular to the paper plane. IGNORing this convention is, well, you can guess it ;-)

ANSWER: We have reviewed the relevant structural figures in the main text, Expanded View, and Appendix to follow the standard orientation convention, presenting the substrate from N- to C-terminus, left to right. This adjustment improves clarity and aligns with widely accepted visualization standards, as suggested.

39. "Fab region and IgA1 HR fused to the Fc region of IgG1" - is NOT (or only partly shown in figure 4D).

ANSWER: We agree with the reviewer that the Fab region and IgA1 HR fused to the Fc region of IgG1 were not clearly represented in the figure (now Figure 3D). We have now added the IgA1Fab_{-HR}-IgG1_{Fc} construct to the figure and revised the numbering of all constructs to provide a more detailed and accurate representation of these antibodies.

40. Line 250. While I like the choice of the IgA-IgG fusion constructs (of which only one expressed), I am sceptical with interpretation that the HR domain alone is insufficient for

substrate recognition. I find it very optimistic to assume that the fused domains would behave neutral. Remember Hippocrates: first, do no harm!

ANSWER: We appreciate the reviewer's comment and agree that fusion constructs should be carefully interpreted, as they can potentially influence structural or functional properties. To address this, we analyzed the behavior of IgA1_{fl}, IgG1_{fl}, and the IgG1_{Fab}-IgA1_{HR-Fc} fusion construct by SEC, as shown in the newly added Appendix Figure S6G.

The SEC profiles of IgA1_{fl} and the IgG1_{Fab}-IgA1_{HR-Fc} display similar elution volumes and peak shapes, which suggests that, at least under the conditions tested, the fusion construct behaves comparably to native IgA1.

Taken together with the lack of activity against the MBP-TEV-IgA1_{HR}-EGFP construct (please, see point 34 of this revision) and a synthetic decapeptide corresponding to the IgA hinge region, these results strongly support the conclusion that the IgA1 hinge alone is insufficient for substrate recognition by ThomasA.

- any additional non-essential suggestions for improving the study (which will be at the author's/editor's discretion)

41. Frankly, I found the manuscript rather difficult to read. Maybe the structure of the paper can be made clearer.

ANSWER: We have reorganized the article's structure to enhance readability and logical flow. Additionally, we have shortened and streamlined the structural comparison between ThomasA and BF3526 with other homologues to make these sections more concise and focused.

Referee #3:

42. I apologize for the tardy submission of the report on this manuscript. I read it back and forth carefully to find any points I could criticize, but I found nothing. Since I'm not an expert in structural biology, I concentrated on experimental design and how the research was executed. From a biological point of view, the manuscript is perfect, and the conclusions are based on results generated by complementary methods such as AF3 modeling, SAXS, and X-ray crystallography. Along with the results of mutagenetic experiments, the manuscript presents an unbiased molecular model of the structure and function of IgA proteases belonging to the M64 family of metalloproteases. The divergent substrate recognition strategies used by two representative proteases are fascinating in the context of the conserved catalytic mechanism. In addition, the paper is very well written and illustrated; it is a true pleasure to read. I hope other reviewers also find this manuscript presents an outstanding piece of work deserving of publication in EMBO J.

ANSWER: We sincerely thank the reviewer for their thoughtful and generous evaluation of our manuscript. We greatly appreciate the recognition of our experimental design and the integration of complementary methods, as well as the positive feedback on the clarity of the writing and illustrations.

REFERENCES

Rambo RP & Tainer JA (2013) Accurate assessment of mass, models and resolution by small-angle scattering. *Nature* 496: 477–481

Snapp EL, Hegde RS, Francolini M, Lombardo F, Colombo S, Pedrazzini E, Borgese N & Lippincott-Schwartz J (2003) Formation of stacked ER cisternae by low affinity protein interactions. *The Journal of Cell Biology* 163: 257–269

Dear Beatriz,

Thank you for submitting your revised manuscript (EMBOJ-2025-120971R) to The EMBO Journal for our consideration, and for your patience during peer review. Your manuscript has been sent back to the original referees #1 and #2, who had previously reviewed the initial version of your manuscript, and we have now received their comments, which you can find below.

I am very pleased to say that both referees are very satisfied with your revision, acknowledge that all initially raised concerns have been adequately addressed, and now support the publication of your manuscript in our journal without any further comments. In light of this expert input, I am glad to inform you that your manuscript has been in principle accepted for publication in The EMBO Journal. Congratulations on an excellent work!

Before we can move forward with formal acceptance and publication of your article, there are a few changes and corrections we need you to make in a final version of your manuscript:

- Please provide your institutional e-mail address for correspondence on the first (title) page of your revised manuscript.
- The funding information mentioned in the Acknowledgements section of your manuscript and that provided in our manuscript tracking system (eJP) should be identical. Currently, the following information is missing in eJP: "Ramón y Cajal" (RYC2020-028922-I/AEI/ 10.13039/501100011033) and "Programa de movilidad del personal investigador doctor (2023)" (MV_2024_1_0005; Basque Government) fellowships.
- Please provide a list of up to 5 keywords (preferably broad terms that would enhance online search engine discoverability of your article) after the Abstract of your revised manuscript.
- Thank you for providing referee access to your deposited data. The tokens for access to the confidential data can now be removed from the Data availability section. Please make sure that all data will be publicly available at the time of publication, and that all full specific URLs to the deposited datasets are included in the Data availability statement.
- The author contributions statement should be removed from the manuscript file. Instead, we use CRediT to specify the contributions of each author in the journal submission system. Please feel free to use the free text box to provide more detailed descriptions during submission. See also our guide to authors for more information: <https://www.embopress.org/page/journal/14602075/authorguide#authorshipguidelines>.
- We noticed that callouts for Fig. 6B, 7A are missing.
- Please note that no answer has been provided in section "Laboratory protocol" of your Author Checklist. Please select "Yes" (and list in which sections of the manuscript the relevant information can be found) or "Not Applicable", and re-upload your completed Author Checklist.
- The heading on the first page of the Appendix PDF file should be "Appendix for", followed by the manuscript's title.
- Please note that EMBO press papers are accompanied online by:
 - A) a short (2 sentences) summary of the findings and their significance,
 - B) 2-5 short bullet points highlighting the key results, and
 - C) a synopsis image in .jpg or .png format that is exactly 550 pixels wide and 300-600 pixels high (the height is variable). Please note that all text needs to be legible at the final size.Please upload this information along with your revised manuscript (the text for A and B should be provided in a separate Word file).
- During our routine pre-publication Figure checks, our team detected high similarity between left and right images of Appendix Fig. S13 - D, E and F. Could you please confirm that this similarity is intentional?
- The order of the manuscript sections must be corrected as follows: Title page - Abstract and Keywords - Introduction - Results - Discussion - Methods - Data Availability - Acknowledgements - Disclosure and Competing Interests Statement - References - Figure Legends - main Tables (if there are any) - Expanded View Figure Legends.

Please also note that as part of the EMBO publications' Transparent Editorial Process, The EMBO Journal publishes online a Peer Review File along with each accepted manuscript. This File will be published in conjunction with your paper and will include the referee reports, your point-by-point response and all pertinent correspondence relating to the manuscript. You can opt out of this by letting the editorial office know (contact@embojournal.org). If you do opt out, the Peer Review File link will point to the following statement: "No Peer Review File is available with this article, as the authors have chosen not to make the review

process public in this case."

We look forward to seeing a final version of your manuscript as soon as possible. Please let us know if you have any questions and use this link to submit your revision: xxxxxxxxxxxxxxxxxxxxxxxxxxxxxxx

Best regards,

Ioannis

Referee #1:

The authors have satisfactorily addressed all points raised by this reviewer. I fully support the publication of this manuscript in The EMBO Journal.

Referee #2:

I find the manuscript now ready for publication.
congratulations to the authors for the fine work!

All editorial and formatting issues were resolved by the authors.

Dear Beatriz,

Congratulations on an excellent work! I am very pleased to inform you that your manuscript has been accepted for publication in The EMBO Journal. Thank you for comprehensively addressing the initially raised referees' concerns and all editorial requests for corrections and changes.

If you have any questions, please do not hesitate to contact the Editorial Office. Thank you for your contribution to The EMBO Journal. Working with you has been a pleasure!

Best regards,

Ioannis
